# Transcriptomic Analysis of *Ficus carica* Peels with a Focus on the Key Genes for Anthocyanin Biosynthesis

**DOI:** 10.3390/ijms21041245

**Published:** 2020-02-13

**Authors:** Jing Li, Yuyan An, Liangju Wang

**Affiliations:** College of Horticulture, Nanjing Agricultural University, Nanjing 210014, China; 2016204002@njau.edu.cn

**Keywords:** fig, anthocyanin, transcriptomic analysis, *FcCHS1*, *FcCHI1*, *FcDFR1*, *FcMYB21*, *FcMYB123*

## Abstract

Fig (*Ficus carica* L.), a deciduous fruit tree of the Moraceae, provides ingredients for human health such as anthocyanins. However, little information is available on its molecular structure. In this study, the fig peels in the yellow (Y) and red (R) stages were used for transcriptomic analyses. Comparing the R with the Y stage, we obtained 6224 differentially expressed genes, specifically, anthocyanin-related genes including five *CHS*, three *CHI*, three *DFR*, three *ANS*, two *UFGT* and seven *R2R3-MYB* genes. Furthermore, three anthocyanin biosynthetic genes, i.e., *FcCHS1*, *FcCHI1* and *FcDFR1*, and two *R2R3-MYB* genes, i.e., *FcMYB21* and *FcMYB123*, were cloned; sequences analysis and their molecular characteristics indicated their important roles in fig anthocyanin biosynthesis. Heterologous expression of *FcMYB21* and *FcMYB123* significantly promoted anthocyanin accumulation in both apple fruits and calli, further suggesting their regulatory roles in fig coloration. These findings provide novel insights into the molecular mechanisms behind fig anthocyanin biosynthesis and coloration, facilitating the genetic improvement of high-anthocyanin cultivars and other horticultural traits in fig fruits.

## 1. Introduction

Fig (*Ficus carica* L.) belongs to the Moraceae. As one of the earliest domesticated crops, it is mainly planted in Mediterranean countries [1]. There are four ecotypes, i.e., caprifig, Smyrna, San Pedro, and common type. Most cultivated varieties of fig are of the common type, bearing a closed syconium fruit without pollination [2]. Fig fruits with green, yellow, and red peels accumulate different amounts of anthocyanins in their peels and drupelets inside the syconia [3]. The red color is mainly dependent on the species and concentration of anthocyanins. Previous reports have indicated that purple fig fruits contain cyanidin-3-o-glucoside, cyanidin-3-rutinoside, pelargonidin-3-glucoside, and cyanidin-3,5-diglucoside [4], while cyanidin was the predominant anthocyanin in the peels of the red cultivar ‘Brown Turkey’ [5]. Anthocyanins are not only an attractive factor of fruit quality, but are also favorable for plant development and defense against environmental stress, and in the prevention heart disease and cancers for human beings [6].

The biosynthetic pathways of anthocyanins have been well studied in vascular plants [7]. Anthocyanin biosynthesis is catalyzed by a series of enzymes. The first section starts with L-phenylalanine. It is sequentially catalyzed by phenylalanine ammonium lyase (PAL), cinnamic acid 4-hydroxylase (C4H), and 4-coumaroyl CoA ligase (4CL) to form cinnamic acid, *p*-coumaroyl CoA, and then 4-coumaroyl CoA, respectively. These steps belong to the phenylpropanoid pathway, and the products serve as the biosynthetic precursors of flavonoids, lignins, etc. The following section enters the flavonoid pathway. Naringenin is formed by the catalysis of chalcone synthase (CHS) and chalcone isomerase (CHI), and is then converted into dihydroflavonols by flavanone 3-hydroxylase (F3H) and flavanone 3’-hydroxylase (F3’H). After that, dihydroflavonols are catalyzed by three enzymes, i.e., dihydroflavonol reductase (DFR), anthocyanidin synthase (ANS), and UDP-glucose: flavonoid 3-o-glucosyltrans-ferase (UFGT) to form anthocyanins. The nine genes mentioned above are expressed differently among plant species, organs, and developmental stages. The downstream genes, especially *DFR*, *ANS,* and *UFGT,* are positively correlated with anthocyanin accumulation [8,9].

Increasing evidence has shown that transcription factors play an indispensable role in modulating color changes [10]. It is well known that the synthesis of anthocyanins is controlled by MYB transcription factors, basic helix-loop-helix (bHLH) proteins, and WD40 proteins at the transcriptional level, particularly R2R3-MYBs. The overexpression of *AtPAP1* (*AtMYB75*) or *AtPAP2* (*AtMYB90*) in purple transgenic tobacco plants strongly enhances anthocyanin contents via upregulating all of the anthocyanin biosynthetic genes [11]. More and more *MYBs* involved in anthocyanin biosynthesis regulation have been identified and characterized in fruit species, e.g., apple and grape. For example, some *R2R3-MYB* genes that are related to anthocyanin regulation in apple, including *MdMYB1* and *MdMYB10*, had been isolated from different apple cultivars; both could promote anthocyanin accumulation [12,13]. *VvMYBA1* and *VvMYBA2* were also reported to display a similar function in grape [14,15]. However, in fig, little information is available on the MYB regulators of anthocyanin biosynthesis.

Fig is becoming more and more popular for its edible value and medicinal properties [16]. In Shandong province of China, its cultivation area had increased from 1000 ha to 3300 ha by the end of 2014 [17]. Although the components of fig anthocyanins have been isolated and distinguished [5], studies about its anthocyanin biosynthesis and regulatory mechanisms are seldom conducted in the field of molecular biology. So far as we know, only *FcANS1* has been cloned, providing sequence information regarding the fig anthocyanin pathway [18]. Therefore, in this study, using fig peels as materials, we performed high-throughput RNA-sequencing (RNA-seq) research to analyze the key biosynthetic and regulatory genes of anthocyanin metabolism. Some key related genes were then cloned and transiently transformed to apple fruits and calli to test their roles in anthocyanin accumulation.

## 2. Results

### 2.1. Anthocyanin Contents in Fig Fruit Peels

During fig fruit development, pigmentation was visually apparent (Figure 1A); this coincided with the abundance of anthocyanins in the peels (Figure 1B). Quite low levels of anthocyanins were detected in the yellow peels, but anthocyanin contents increased rapidly, by 28 times, within the following five days (Figure 1C). Based on the peel colors of fig fruits, uncolored (yellow, Y) and colored fruits (red, R) were selected for subsequent experiments.

### 2.2. Library Construction, De novo Assembly, and Annotation

In order to identify anthocyanin-related genes, Y and R libraries were constructed. The total number of raw reads generated in each library ranged from 48,269,776 to 55,542,173. After filtering the raw data, the total number of clean reads per library was from 45.35 to 52.78 million with a Q20 percentage of over 97% and a GC percentage of about 47% (Appendix A). These results suggested that the sequencing data were sufficient to ensure accurate de novo assembly. As a result, 269,684 transcripts were obtained in total, with a mean length of 1126 bp and an N50 of 2014 bp. Among them, we obtained 186,048 unigenes, with an average size of 1498 bp and an N50 of 2172 bp. The sequences, ranging from 200 bp to 2000 bp in length, were divided into four groups; 102,861 unigenes (55.29%) were up to 1000 bp in length (Appendix A).

All 186,048 unigenes were blasted against seven public databases; 154,920 were matched homologous sequences in at least one database (Appendix A). Among the unigenes, 74.27% (138,194) sequences were mapped to known proteins in the NCBI nonredundant protein sequences (Nr) database, and the mapping ratio was 14.99% (27,893) in all databases.

The smaller E values indicated higher similarity to available plant sequences. In total, 68.20% of the mapped genes in Nr database showed E-value < 10^−45^ (Appendix A). Moreover, 42.70% and 43.90% sequences presented 80–95% and 60–80% similarity, respectively, with known sequences (Appendix A). About 15.60% unigenes had the best blast hits to sequences from *Prunus mume*, followed by 12.50% to *Prunus persica* sequences (Appendix A).

### 2.3. Analysis of Differentially Expressed Genes

Compared to the Y library, a total of 6224 differentially expressed genes (DEGs, Appendix A) were identified in the R library, including 58.15% upregulated and 41.85% downregulated genes (Figure 2A). To obtain a comprehensive understanding of DEGs, gene ontology (GO) and Kyoto encyclopedia of genes and genomes (KEGG) -based functional enrichment were conducted. According to GO assignments, 4566 DEGs were divided into three main categories: biological process, cellular component, and molecular function (Appendix A). However, the top 30 enriched items did not cover cellular components (Appendix A). The dominant subcategories within the biological process category were ‘metabolic process’ (63.12%) and ‘single-organism process’ (50.64%). Among the molecular function category, ‘catalytic activity’ (2691, 59.94%) and ‘ion binding’ (1612, 35.30%) were the two most abundant classes.

Among the KEGG pathway analysis, biosynthesis of secondary metabolites such as ‘flavonoid biosynthesis’ (ko00941), ‘flavone and flavonol biosynthesis’ (ko00944), and ‘phenylpropanoid biosynthesis’ (ko00940) represented the top fifteen enriched KEGG pathways (Figure 2B, Appendix A). Notably, ‘flavonoid biosynthesis’ was the dominant pathway, containing 74 DEGs.

To confirm the reliability of transcriptomic data, twelve DEGs related to anthocyanin were randomly selected for quantitative real-time PCR (qRT-PCR, primer sequences are shown in Appendix A). The qRT-PCR data were generally consistent with the RNA-seq results (Appendix A), indicating the effective quality of our RNA-seq data.

### 2.4. DEGs Related to Anthocyanin Biosynthesis

There were 152 anthocyanin related unigenes. Among them, 73 DEGs were identified; there were 26 DEGs left after excluding 47 short fragments of *CHS*. All these DEGs exhibited high expression in R stages (Figure 3). There were six genes belonging to the phenylpropanoid pathway (two *PALs*, one *C4H*, three *4CLs*). Except for one of *4CLs* (Cluster-11055.11791), which was upregulated 5.16 fold, the other five genes were all upregulated by 2.52–3.00 times in R compared to Y. Only one *F3H* and three *F3’Hs* were significantly upregulated, and they shared the same expression trend as the five genes in phenylpropanoid pathway, except for one *F3’H* (Cluster-11055.82169), which showed the same trend with *4CL* (Cluster-11055.63898). The fragments per kilobase of gene per million mapped reads (FPKM) values of *CHS* (Cluster-11055.81810) and *CHI* (Cluster-11055. 79888) were above 1000 in red stage; these two genes showed 3.19- and 3.90-fold upregulation in R vs. Y. Two *CHS* genes (Cluster-11055.103287 and Cluster-11055.148622) and one *CHI* (Cluster-11055.10763) showed about 32 and 16 fold upregulation, respectively. One *DFR* (Cluster-11055.59175) showed the highest FPKM value compared with the other two (Cluster-11055.59172 and Cluster-11055. 146588); it was upregulated 3.12 times. Three *ANSs* and two *UFGTs* showed about 7-fold increments during the transition from yellow to red.

### 2.5. Candidate MYBs of Anthocyanin Biosynthesis

In DEGs, 19 members of the *MYB* family were filtrated (16 upregulated and 3 downregulated). Prediction of the conserved motif by SMART showed that seven genes contained completed R2 and R3 motifs. Moreover, these seven genes showed increasing expression from the uncolored to the colored stages. Compared with R, the expression level of Cluster-11055.82418 could not be detected in Y. Cluster-11055.69229 possessed the highest FPKM value among the seven genes, and showed 1.55-fold upregulation. 

A phylogenetic tree was constructed using 126 R2R3-MYB proteins from *Arabidopsis* and seven MYBs from DEGs (Figure 4A). Cluster-11055.86015 was closely related to AtMYB123, and *TT2*-type genes were involved in anthocyanin and proanthocyanidin biosynthesis regulation [19,20]. Cluster-11055.86015 composed a full-length open reading frame (ORF), and therefore, was named *FcMYB123*. Cluster-11055.69229 was in the branch containing AtMYB21 and AtMYB24, which were found to regulate male fertility or flavonoid biosynthesis [21,22]. Cluster-11055.69229 shared 59.05% and 58.77% amino acid identities with AtMYB21 and AtMYB24, respectively. We named it *FcMYB21*. The other five MYB proteins were members of different branches.

### 2.6. Molecular Characteristics and Sequence Analysis of FcCHS1, FcCHI1 and FcDFR1

According to the sequences data of transcriptomic analysis on anthocyanin biosynthesis, *FcCHS1*, *FcCHI1,* and *FcDFR1* were cloned from fig red peels (primer sequences were shown in Appendix A). The full length ORF of *FcCHS1* coded 389 amino acids, the polypeptide with a calculated molecular weight (Mw) of 42.66 kDa and a theoretical isoelectric point (pI) of 6.12. *FcCHI1* encoded a polypeptide of 257 amino acids, with an Mw of 28.14 kDa and pI of 6.15 (Appendix A). The completed amino acid sequence encoded by *FcDFR1* was 363 in length. Those three genes were all hydrophilic protein. Their secondary structures all contained α-helices, random coils, β-turns, and extended strands. In addition, α-helices were evenly arranged throughout each sequence, and random coils ranged from 31.91% to 38.24%. β-turns were the least abundant of the four structures, and varied from 7.97% to 9.34%. Comparing them with their orthologs whose function for anthocyanin biosynthesis have been proved, we found that they were all acidic amino acid and hydrophilic protein. Their proportion of secondary structure were extremely similar, and the variation was less than 4%.

Several conserved active sites (Cys 163, His 303 and Asn 336) were found in FcCHS1, which were marked with red asterisks above the ten sequences (Figure 5A). Two phenylalanine (F) residues which played a role in substrate specificity were observed. Analysis of the phylogenetic tree showed that CHSs were clustered into three group, where FcCHS1, MnCHS1, and MnCHS2 (*Morus notabilis*) were in the same branch. Also, FcCHS1 showed a close relation with the CHSs from the Rosaceae plants (Figure 5D). Those CHSs proteins of mosses and Lycophyte were regarded as an outgroup. Confirmed FcCHS1 are orthologs of known CHS proteins. And FcCHS1 shared 94.86% similarity with MnCHS1 and MnCHS2.

The proposed amino acid sequences of FcCHI1 shared four conserved residues in Thr49, Tyr107, Asn114, and Ser191 with those of other plant species (Figure 5C), which are essential for CHI activity. According to the two residues (red dots, Figure 5C) involved in catalytic residues, FcCHI1 likely belongs to type-I. A phylogenetic tree created by the neighbor-joining method showed that the FcCHI1 protein was closely related to MnCHI of *Morus notabilis* (Figure 5F). Moreover, orthologs from liverworts rooted FcCHI1 to CHIs family. The deduced FcCHI1 showed 60.85%, 58.91%, and 57.36% identities with PaCHI (*Prunus avium*), MdCHI (*Malus* × *domestica*), and PmCHI (*Prunus mume*), respectively. 

FcDFR1 possessed a high degree of similarity to the DFRs of other plants, such as *Morus notabilis* (76.76%), *Durio zibethinus* (75.57%), and *Prunus persica* (75.14%). The FcDFR1 sequence possessed a NADP binding region at the N-terminal including 21 residues. In addition, TVNVEPQTKFFYDESCWSDIQFCRTV positioned at 129–156 were substrate binding regions. The asparagine (N132) represents the key residue to the determination of substrate specificity of FcDFR1 (Figure 5B). In the phylogenetic tree, BEN1 proteins (BRI1-5 ENHANCED 1, DFR like protein) as an outgroup, verified that FcDFR1 are typical DFRs. FcDFR1and MnDFR were in the same clade (Figure 5E). 

### 2.7. Sequence Characterization of FcMYB21 and FcMYB123

Two full-length gene sequences encoded R2R3 MYBs were isolated based on fig RNA-seq (primer sequences were shown in Appendix A). Both FcMYB21 and FcMYB123 composed distinct R2 and R3 repeat domains. Each R repeated fragment consisted of 51–52 conserved amino acid residues (Figure 4B,C).

The full-length ORF of *FcMYB21* and *FcMYB123* were 684 bp and 1008 bp, respectively. The Mw of FcMYB21 and FcMYB123 were 26.14 kDa and 37.37 kDa, respectively. The predicted pI of the two genes were 7.92 and 5.40, respectively. Random coils accounted for the largest proportion of FcMYB21 (49.78%) and FcMYB123 (60.90%) among the putative secondary structures (Appendix A). FcMYB21 and FcMYB123 presented a similar secondary structure with AmMYB305 and MdMYB9, respectively, the two MYBs that have been shown to regulate anthocyanin accumulation.

### 2.8. The Function of FcMYB21 and FcMYB123 in Apple Peels and Calli

Transient assays with apple peels and calli were performed to test the function of *FcMYB21* and *FcMYB123*. The *Agrobacterium* strain with *FcMYB21* or *FcMYB123* was injected into apple peels, and the empty plasmid was used as a control. Both 35s::*FcMYB21* and 35s::*FcMYB123* significantly promoted anthocyanin accumulation in apple peels and upregulated the expression levels of apple anthocyanin biosynthesis genes including *MdCHS*, *MdCHI*, *MdDFR*, *MdANS*, and *MdUFGT* (Figure 6A–C,F,G).

To verify this injection assay, transgenic apple calli was used to validate the function of *FcMYB21* and *FcMYB123*. Compared with the control, both *FcMYB21*- and *FcMYB123*- overexpressed calli looked redder and accumulated markedly higher level of anthocyanins (Figure 6D,E). Heterologous expression of *FcMYB21* and *FcMYB123* also upregulated the expression of anthocyanin-related genes in apple calli (Figure 6H,I). These results suggest that *FcMYB21* and *FcMYB123* may play important roles in anthocyanin biosynthesis regulation.

The expression levels of *MdMYB9* and *MdMYB11* in *FcMYB21*- and *FcMYB123*-overexpressed apple peels and calli decreased or did not change in comparison with wild type. In contrast, the transcription level of *MdMYC2* and *MdbHLH3* increased sharply in *FcMYB21*- and *FcMYB123*-overexpressed apple peels and calli. These results indicated that the effect of FcMYB21 and FcMYB123 may be independent of endogenous MdMYB9 and MdMYB11, but related to bHLH partners. 

## 3. Discussion

### 3.1. De novo Assembled the Peels Transcriptome during Coloration

Lacking gene sequence information limited genetic research of fig coloration. Next generation sequencing technology is an effective and common method by which to obtain massive sequences, especially for plants without a reference genome [23]. To our knowledge, only *FcANS1* was cloned and given the sequence information in the fig anthocyanin pathway [18]. To maximize the DEGs, we sequenced two stages of figs with large differences in coloration. It is easy to distinguish the turning stage of figs by color because fruits in this stage are yellow instead of dark green. According to previous reports [8,24], DEGs of anthocyanin biosynthesis between the yellow and red stage were almost the same as those in the green and red stage because those genes were not significantly different between the green and yellow stages. Therefore, the yellow and red stages were selected to do the transcriptomic analysis. In total, we isolated 186,048 unigenes from the two libraries with an average length of 1498 bp. In previous reports on figs, the average length was 683 bp and 302 bp, respectively [25,26]. Also, the average length was around twice as long as that of litch (737 bp) [8]. Whole genome sequences of ‘Horaishi’ were taken in 2017 through the Illumina platform in conjunction with the shotgun method [27], but the data have not published yet. 

Based on the threshold of Foldchange >2 and false discovery rate <0.05, we separated 6224 genes. As expected, the amount of DEGs obtained from RNA-seq was almost 60 times larger than that of suppression subtractive hybridization. In total, only 104 DEGs from ‘Fuji’ peels and 93 DEGs from litchi floral bud were generated by using suppression subtractive hybridization [28,29]. Overall, the quality of our transcriptome were sufficient for subsequent analysis.

### 3.2. Candidate and Cloned Structural Genes of Fig Peels

Fig is a kind of anthocyanin-rich fruit. The main purpose of this article was to analyze key enzymes and regulatory factors involved in the anthocyanin accumulation of fig peels. In the DEGs, we identified the genes that took part in each step of anthocyanin biosynthesis. *PAL*, *4CL,* and *C4H* are all common multigene families, usually consisting one to five members. PAL is the first enzyme of the phenylpropanoid pathway, and is considered a rate-limiting enzyme [30]. There were four *PALs* in *Arabidopsis*, five in polar, and three in *Coffea canephora* [31,32,33]. About four, four, and five members of *4CLs* were found in *Arabidopsis*, soybean, and rice, respectively [34,35,36]. Here, we found that two *PALs* and three *4CLs* were upregulated, indicating that there are more than one gene in the fig *PAL* and *4CL* family. In comparison with wild-type *Arabidopsis*, the deletion mutant of *At4CL3* reduced approximately 80% of the flavonol glycoside contents [34]. *Mn4CL3* has been predicted to be associated with flavonoid biosynthesis [37]. In this study, the amino acid sequence of Cluster-11055.81897 (4CL) showed a close relationship with and Mn4CL3, and therefore, was considered as one of the candidate genes for pigmentation. *Arabidopsis* and parsley have been reported to harbor only one *C4H* gene [38,39], which affects anthocyanin and ligin biosynthesis. Similarly, here, only one *C4H* gene was found in our upregulated gene pool, indicating that it may contribute to anthocyanin accumulation in fig peels. We obtained five *CHS*, three *CHI*, one *F3H,* and three *F3’H* genes in generating dihydroflavonols, and they kept high expression levels in the red stage. CHS is regarded as a rate-limiting enzyme in catalyzing 4-coumaroyl CoA with malonyl CoA to form 2’,4’,6’,4-tetrahydroxy-chalcone (naringenin chalcone), and it is also a multigene family [40]. In our report, we found five *CHSs* with lengths over 700 bp; there were another 47 *CHS* genes with about 300 bp in length. A large number of short sequences may represent different fragments of one gene. The SMART data further indicated that those short *CHS* unigenes did not possess complete conservative domains. Therefore, these short sequences were not listed as valid genes. CHIs catalyze the second reaction after CHSs. There are two types CHIs in plants. Type I CHIs are ubiquitous, while type II mainly appear in legumes [41]. The three CHIs in our DEGs (Cluster-11055.10763, Cluster-11055.145351, and Cluster-11055.79888) were all members of type I, which isomerized 6′-hydroxychalcone to 5-hydroxyflavanone in forming primary C15 flavonoid skeleton [42]. *F3H* and *F3’H* have received attention. The seed coat, leaf, and stem of *Arabidopsis F3H* mutant (*tt6*) showed lower anthocyanin contents than the organs of the wild-type [43]. The seed coat of *tt7* mutant (*AtF3’H*) turned ivory from brown in wild type [44]. The blast best hit *Arabidopsis thaliana* genes of *F3H* (Cluster-11055.81147) and *F3’H* (Cluster-11055.79806) were *AtTT6* and *AtTT7*, respectively, suggesting their possible roles in anthocyanin biosynthesis.

The transcript levels of *DFR*, *ANS,* and *UFGT* showed a high correlation with the anthocyanin content in many fruits, such as apple, pear, and strawberry [45,46,47]. Here, we show similar results in Figure 3, i.e., three *DFRs*, three *ANSs*, and two *UFGTs* in our DEGs were all upregulated. The OFR of one *ANS* (Cluster-11055.79498) was exactly the same as *FcANS1* (did not upload to NCBI), which was cloned from ‘Brown Turkey’ and participated in anthocyanin biosynthesis [18]. *UFGT* genes are regarded as a key control point in anthocyanin biosynthesis. As in earlier reports, its transcript level and duration time always influenced anthocyanin content [48]. In the present study, the expression level of Cluster-11055.59183 and Cluster-11055.59184 sharply increased in parallel with the accumulation of anthocyanin contents during fruit pigmentation. Although both demonstrated a close relationship with MaUFGT2, a putative major gene in regulation anthocyanin synthesis in *Morus alba* [49], the predicted polypeptide sequence of Cluster-11055.59183 mapped a higher similarity to MaUFGT2, indicating that Cluster-11055.59183 may play the similar role in fig.

Wang et al. [50] declared that they cloned six structural genes for fig anthocyanin biosynthesis, i.e., *FcCHS*, *FcCHI*, *FcF3H*, *FcF3’H*, *FcDFR,* and *FcUFGT,* and that those genes can be induced by light, suggesting their roles in anthocyanin accumulation. However, no sequence information was found in NCBI and no bioinformatics analysis were performed in that study. Here, we did not focus on light response, but rather, on fig development. We cloned *FcCHS1*, *FcCHI1,* and *FcDFR1,* along with the fig peels changing from yellow to red. Moreover, to better understand their roles in fig anthocyanin biosynthesis, detailed sequence analyses and molecular characteristics were taken out. The similar molecular characteristics and proportion of the secondary structures of FcCHS1, FcCHI1, FcDFR1 with their orthologs proteins whose function have been proved in anthocyanin accumulation imply their role in fig coloring (Appendix A). The vital role of the *CHS* and *CHI* have been verified in massive plants by phenotypic and metabolic methods on overexpression lines or deleted mutants. The suppression of the *CHS* gene in dahlia and gentian caused white flowers [51,52]. A frameshift in coding region of *CHI* resulted in gold onions [53]. 

*DFRs* are also a key point site in anthocyanin biosynthesis. Torenia, with antisense *DFR* gene, produced bluer flowers than wild type by accumulating more flavones [54]. According to the amino acid residues at 134, *DFR* genes are divided into three categories: Asn-type (asparagine, N), Asp-type (Asparticacid, D), and non-Asn/Asp type [55]. Asn-type is the most common type, and the non-Asn/Asp type exists in very few species [55]. But the position of the decisive amino acid changes with the position of substrate binding region. For example, Asn of GbDFR3 was posited at 124 and the amino acids MnDFR (Asp) and GbDFR1 (Lysine, K) were at 132 and 154, respectively [56,57]. The crucial amino acid residues directly affect the substrates of DFR enzymes. DFRs utilize dihydrokaempferol, dihydromyricetin, or dihydroquercetin as substrates to produce unstable leucopelargonidin, leucocyanidin, and leucodelphinidin [57]. DFRs of *petunia* and *cymbidium hybrida* were Asp-type. they could not catalyze dihydrokaempferol, resulting in the inability to accumulate orange pelargonidin in petal [55,58]. FcDFR1 is a member of Asn-type; therefore, the variety of substrates lays a foundation for various anthocyanins and peel colors in figs. Throughout the multiple alignments, the entire coding region of FcDFR1 harbored strong identities with DFRs from other plants. FcCHS1, FcCHI1, and FcDFR1 presented the same conserved amino acid residues as the corresponding genes of other vascular plants [55,56,59] and their phylogenetic analysis supported their close relationship with CHSs, CHIs, and DFRs of other species. These results indicate that they are truly the ortholog of known proteins in the three families. Moreover, FcCHS1, FcCHI1, and FcDFR1 showed strong similarities and had a close relationship with counterparts of *Morus notabilis*. MnCHS1, MnCHS2, MnCHI, and MnDFR1 have been shown to be involved in anthocyanin accumulation in the flowers of overexpressing tobacco lines [56]. The three cloned genes may perform the same function in fig coloration. The three proteins of fig all grouped with proteins from the Rosaceae. This information also help us to predict the gene function and trace the evolution process.

### 3.3. Candidate and Validated Functions of MYBs of Fig Peels

*MYB* transcription factor is one of the largest gene families in plants, acting as a central regulator in growth, development, and stress responses [60]. *AtMYB14* plays an important role in freezing tolerance of *Arabidopsis* [61]. *AtMYB108* was a redundant gene with *AtMYB24* in controlling anther development [62]. At the same time, *AtMYB44* was a candidate gene in ABA-mediated stress responses [63]. Here, we identified seven *R2R3-MYB* genes (Appendix A). Among them, Cluster-11055.106720 and Cluster-11055.107006 were similar to AtMYB14. Cluster-11055. 93035 was clustered with AtMYB108, and Cluster-11055. 82418 was in the same branch with AtMYB44. These results suggest some potential regulators for fig development and stress responses.

Another role of MYB transcription factors is regulating fruit and flower color formation [10]. The function of R2R3-MYBs has been well studied in *Arabidopsis* [64], especially *AtPAP1* and *AtPAP2* in subgroup 6 [65]. The transcriptomics of fig peels taken by Wang et al. [25] showed that the protein sequence of c43569_g1 harbored a high identity (72%) to MdMYB110a, which was a key PAP1-type protein regulating red flesh apple phenotype [66]. However, in this study, we did not find a *MYB* ortholog gene with *AtPAP1*. Moreover, the genome-wide analysis of mulberry MYB transcription factors concluded that no MYBs were classified into subgroup 6 [67]. Therefore, the question of whether *PAP1*-type genes are present in fig needs further study. *AtTT2*, another *R2R3-MYB* transcription factor, has been reported to enhance proanthocyanidin accumulation, but not anthocyanin accumulation, in seed coat of *Arabidopsis* [20]. However, the overexpression of *MdMYB9* which had a close relationship with *AtTT2*, significantly promoted the content of anthocyanins in apple calli [19]. Here, FcMYB123 harbors a high level of similarity in secondary structure with MdMYB9, indicating that they may participate in regulating the anthocyanin biosynthesis of fig.

In this study, we identified two R2R3-MYB proteins and named them FcMYB21 and FcMYB123, respectively, according to the phylogenetic analysis with 126 R2R3-MYBs of *Arabidopsis*. The derived amino acid sequence of FcMYB21 exhibited a high level of similarity to AtMYB21 proteins, and was shown to have a close relationship with MYB305 [68]. FcMYB21 comprised 64.47% similarity with AmMYB305 and a similar secondary structure, which has been shown to prompt flavonoid biosynthesis [22]. *FcMYB21* may act in the same manner in secondary metabolism. FcMYB123 had a close relationship with AtTT2. We cloned it and performed further analyses. FcMYB123 was indeed a TT2-type regulator. Although AtTT2 was previously reported to regulate proanthocyanin [20]. An et al. [19] recently found that the overexpression of *MdMYB9* or *MdMYB11*, two orthologs of *AtTT2*, enhanced anthocyanin contents in apple calli. The *TT2*-like gene of peach were well shown to perform the same function in promoting anthocyanin accumulation [69]. Therefore, *FcMYB123* may also play a role in anthocyanin accumulation.

The *Agrobacterium* mediated transient transformation of fig fruits and calli are hard to determine because it is much more difficult to separate the flesh and peel of fig fruit. At the same time, gene transient transformation of fig calli does not work well due to the calli browning rapidly, i.e., within 12 h. The transformation system in apple peels and calli have emerged as rapid and effective methods to study gene overexpression and silencing [28,70]. To test the function of *FcMYB21* and *FcMYB123*, in the present study, we conducted transient expression assays using apple peels and calli. The results showed that the overexpression of *FcMYB21* and *FcMYB123* significantly increased the anthocyanin content in both apple skins and calli. The downregulation of *MdMYB9* and *MdMYB11* in transgenic apple peels and calli was similar to the result of overexpression of *LcMYB1* in tobacco, which induced anthocyanin accumulation in flower but repressed the expression of *NtAn2*, an anthocyanin related *MYB* gene, in flower [71]. The results indicate that the effect of *FcMYB21* and *FcMYB123* on anthocyanin biosynthesis may be independent of endogenous *MdMYB9* and *MdMYB11*. The *bHLH* partners *MdMYC2* and *MdbHLH3* showed significantly upregulated in overexpression of *FcMYB21* and *FcMYB123*, respectively. The result was similar with *MrMYB1*. Overexpression of *MrMYB1* in *Arabidopsis* induced anthocyanin accumulation in leaves, and *AtTT8* was significant upregulated [72]. Several studies revealed that bHLH transcription factor AtMYC2 interacted with AtMYB21 and further promoted AtMYB21′s role in fertility [73]. Similarly, in apple fruit and calli, overexpressed *FcMYB21* promoted the expression level of *MdMYC2*. We speculate that FcMYB21 promotes fig fruit anthocyanin accumulation, and that its role may have a close relationship with FcMYC2. Moreover, *MdANS1* and *MdbHLH3* increased dramatically in both apple fruits and calli. FcMYB123 could interact with bHLH transcription factors and might be mainly recruited to the promoter regions of *FcANS1*. The knowledge of *FcMYB21* and *FcMYB123* genes affect fig coloring require further research.

Based on RNA-seq, we cloned three structural genes, *FcCHS1*, *FcCHI1*, and *FcDFR1*, which are candidate genes in anthocyanin biosynthesis. Furthermore, transient expression of two *R2R3-MYBs* in apple demonstrated their regulatory roles in anthocyanin accumulation. Although more verification about their function in fig fruits needs be further conducted, our data provide some molecular bases for fig anthocyanin biosynthesis.

## 4. Materials and Methods 

### 4.1. Plant Materials 

The fruits of fig (*Ficus carica* L. cv. ‘Bojihong’) used in this study were harvested from 7-year-old trees grown in an economical orchard at Yujiawan (32°19′N, 118°60′E), Nanjing, China. Fruit peels were collected at about 65 days (yellow stage, the beginning of the turning stage) and 70 days (red stage when the fruit just developed to the marketable mature stage) after fruit setting (Figure 1A). For each biological replication, 30 fruits at each developmental stage were collected from different trees. Fruit peels on the middle site were peeled with a scalpel after cleaning and immediately frozen in liquid nitrogen and stored at −80 °C until use. Here, fruit peels refer to the tissues outside the receptacle. Three biological replications were conducted. 

The apples (*Malus* × *domestica* Borkh. cv. ‘Fuji’) used in this work were harvested in Taigu (37°23′ N, 112°32′ E), Shanxi Province, China. They were bagged at the middle developmental stage and collected one week before the commercial removal of bags. The harvested fruits were transferred to our lab and kept at 22 °C in the dark until use.

Apple calli were induced from ‘Fuji’ flesh, grown on Murashige and Skoog medium (MS) containing 4.5 mM 2,4-dichloropheno-xyaceticacid and 4.4 mM 6-benzylaminopurine at 22 °C under constant darkness [28]. Twenty five days later, the calli were used for transient expression analysis.

### 4.2. Determination of Anthocyanin Contents 

Total anthocyanin contents were measured according to Zheng et al. [74]. The fig peels, apple peels, and apple calli were taken and incubated in 1% (*v*/*v*) HCl–methanol for 24 h at room temperature in the dark. The absorbance of supernatant was measured at 530, 620, and 650 nm with a spectrophotometer. The anthocyanin contents were calculated using the following formula [75]: content (nmol g^−1^FW) = [(OD530 − OD620) − 0.1 × (OD650 − OD620)]/ε × V/M × 10^6^, V-the liquid volume of extraction; M-the weight of plant tissues, absorbency index of total anthocyanin is 4.62× 10^4^. Mean values were obtained from three independent replicates.

### 4.3. Library Construction, Sequence Assembly and Gene Annotation 

Fig peels of yellow and red fruits were used to build two transcriptome libraries, namely Y and R, respectively. Each sample was tested using three replications. Total RNA was extracted using the RNAprep Pure Plant Kit (TIANGEN, Beijing, China). About 1.5 μg RNA per sample was prepared for high-throughput sequencing. The purified and intact mRNA was enriched using poly-T oligo-attached magnetic beads. Then, the product was broken into short segments of 250–350 bp. After that, the double-stranded cDNA was synthesized and selected with AMPure XP system (Beckman coulter, California, US). PCR amplification was performed and the product were purified by AMPure XP beads. Finally, a library quality assessment was conducted on the Agilent Bioanalyzer 2100 system and the library was sequenced on Illumina Hiseq platform with generating paired-end reads. The data was submitted to the NCBI SRA database; the accession number is SRP246224.

The raw data were processed through in-house perl scripts. Clean reads were obtained after removing reads with an adapter, i.e., reads composing ambiguous ‘N’ bases more than 10% and low quality reads (Qphred ≤ 20 bases) from raw data. All the downstream analyses were based on clean data with high quality. De novo assembly of the transcriptome was performed to gain transcript sequences using the Trinity software with default parameters and no reference sequence [76]. The transcripts were clustered by hierarchical cluster analysis [77] and then the longest transcript sequence was selected from each cluster as unigene. 

To identify putative functions of fig unigenes, we performed blastx against the Nr, the NCBI nonredundant nucleotide sequences (Nt), clusters of orthologous groups of proteins (KOG), and Swiss-Prot with E-value < 10^−5^. The protein family (Pfam) was carried out by HMMER with E-value < 10^−2^. KEGG was taken by KEGG automatic annotation server with E-value < 10^−10^ and GO database was based on Nr and Pfam by Blast2GO v2.5 (E-value < 10^−6^) [78].

### 4.4. Differentially Expressed Genes Analysis 

Gene expression levels of each sample were estimated by RSEM [79]. The input data for DEGs analysis was the readcount values. FPKM was used to present the expression level of genes for FPKMs to normalize the abundances of gene expression [80]. Differential expression analyses of two development stages (contains three biological replicates) were performed using the DESeq2 [81]. An adjusted *p*-value < 0.05 and |log_2_ (foldchange)| > 1 were set as the threshold for DEGs.

### 4.5. Sequences Alignment, Phylogenetic Relationship and Bioinformatics Analysis 

The ORF of nucleotide sequences were carried out online (https://ncbiinsights.ncbi.nlm.nih.gov/tag/orffinder/). The primary structure and physiochemical properties of the deduced amino acid sequences were determined utilizing the Expasy (https://www.expasy.org/). The secondary structure of the predicted protein was analyzed via SOPMA (https://npsaprabi.ibcp.fr/cgi-bin/secpredsopma.pl). MYB protein sequences of *Arabidopsis* were obtained from the TAIR database (https://www.arabidopsis.org/). The protein sequences of other plant species were obtained from the NCBI database (https://www.ncbi.nlm.nih.gov/). The multiple protein sequences alignment was performed using DNAMAN 6.0 software and the conserved domains were analyzed using SMART (http://smart.embl-heidelberg.de/). The phylogenetic trees were conducted using the MEGA 5.0 software with bootstrap analysis with 1000 replicates [82].

### 4.6. Gene Clone and Construct Expression Vectors 

*FcCHS1*, *FcCHI1*, *FcDFR1*, *FcMYB21,* and *FcMYB123* were cloned based on the putative ORFs of fig unigenes from our RNA-seq data. The specific primer pairs (Appendix A) were designed to amplify the ORF sequences using KOD DNA polymerase (TOYOBO, Osaka, Japan) from the cDNA of red fig peels. The reaction conditions were as follows: 96 °C for 5 min, followed by 30 cycles of 94 °C for 30 s, 60 °C for 30 s, 72 °C for 1 min, with 10 min extension at 72 °C. The PCR products were cloned into the pMD19-T simple vectors (TaKaRa, Shiga, Japan). Afterwards, those T-vectors were transferred into DH5α competent cells (Transgen Biotech, Beijing, China) for amplification, and the products were sequenced by GENEWIZ (New Jersey, USA).

The overexpression vectors of *FcMYB21* and *FcMYB123* were constructed via linking their ORFs into the linearized plant transformation vector pBI121 using fast-digest restriction enzymes of *Xba*I and *Bam*HI (Thermo Scientific, Waltham, USA) and T4 ligase (Transgen Biotech, Beijing, China). Then the 35S::*FcMYB21* and 35S::*FcMYB123* recombinant vectors were transformed into *Agrobacterium tumefaciens* EHA105 competent cells, respectively.

### 4.7. Agrobacterium-Mediated Transformation System of Apple Peels and Apple Fruit Calli 

The 35S::*FcMYB21*, 35S::*FcMYB123* and the negative control pBI121 were prepared for injecting and infecting apple calli. The *Agrobacterium* strains were incubated in LB broth with antibiotics at 28 °C and resuspended to OD600 of 0.5 in buffer containing 10 mM MgCl_2_, 10 mM 2-(4-Morpholino) ethanesulfonic acid and 120 μM acetosyringone [83]. The suspension was injected into the middle part of apple fruits and 200 μL of suspension into each point. The transformed apples were kept in the dark for 48 h and then placed in a phytotron under constant 100 μmol m^−2^ s^−1^ photon flux density. After five days, the apple peels were collected for anthocyanin contents and RNA extraction.

The *Agrobacterium* strains carrying the overexpression vectors and pBI121 were resuspended by liquid MS medium to OD600 of 0.5. The 25 day-old apple calli were soaked in the *Agrobacterium* solution at room temperature for 20 min and cocultured on MS solid medium without antibiotics in the dark at 22 °C for 48 h. Then, the calli were washed three times using sterilized distilled water, transferred to screening medium with 85.8 mM kanamycin and 660.7 mM carbenicillin, and placed at 22 °C under continuous 100 μmol m^−2^ s^−1^ photon flux density for 48 h after transgenic selection. The anthocyanin contents of calli were measured according to the method described above [74]. *FcMYB21* and *FcMYB123* overexpressed in apple calli were confirmed by qRT-PCR amplification. All assays were replicated at least three times.

### 4.8. RNA Extraction and Gene Expression Analysis 

Total RNA was isolated from fig peels using the cetyltrimethylam-monium bromide method [84] with slight modification. The total RNA of apple peels and apple calli were extracted by RNA plant plus (TIANGEN, Beijing, China) following the manufacturer’s instructions. The first strand of cDNA was synthesized using the TransScript one-step gDNA removal and cDNA synthesis supermix (Transgen, Beijing, China). The qRT-PCR reaction system was performed utilizing ChamQ SYBR qPCR master mix (Vazyme, Nanjing, China) and each sample contained three replicates. The relative expressions of genes were calculated using the 2^−ΔΔCT^ method. 

### 4.9. Statistical Analysis 

Statistical analysis including variance and significant difference were conducted using SPSS 20.0. Differences at *p* < 0.05 were considered significant.

## Figures and Tables

**Figure 1 ijms-21-01245-f001:**
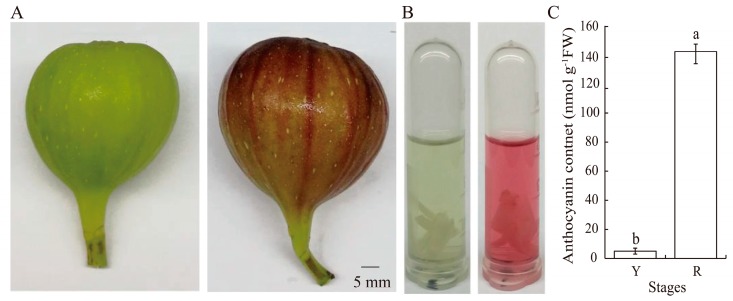
Fig coloration and the anthocyanin contents in yellow and red stages. (**A**) Fig fruits in Y and R stages. Scale bar = 5 mm. (**B**) The colors of anthocyanin extraction in Y and R stages. (**C**) The anthocyanin contents in Y and R stages. Y and R represent yellow (the beginning of the turning stage) and red (the stage when the fruit just developed to the marketable mature stage) stages in Figure (**A**–**C**), respectively. The different small letters represent significant differences (*p* < 0.05).

**Figure 2 ijms-21-01245-f002:**
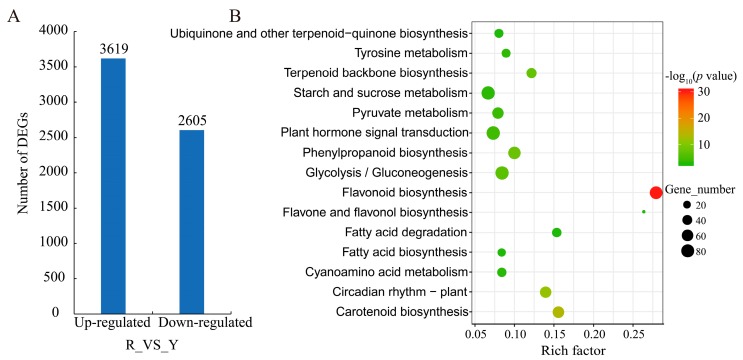
The number and KEGG classification of DEGs. (**A**) DEGs between R and Y stages. R and Y represent fig fruits in the yellow and red stages, respectively. (**B**) The top 15 enriched KEGG pathways of DEGs. The rich factor indicates the ratio of DEGs to the total number of genes in the same pathway. DEGs, differentially expressed genes; KEGG, Kyoto encyclopedia of genes and genomes.

**Figure 3 ijms-21-01245-f003:**
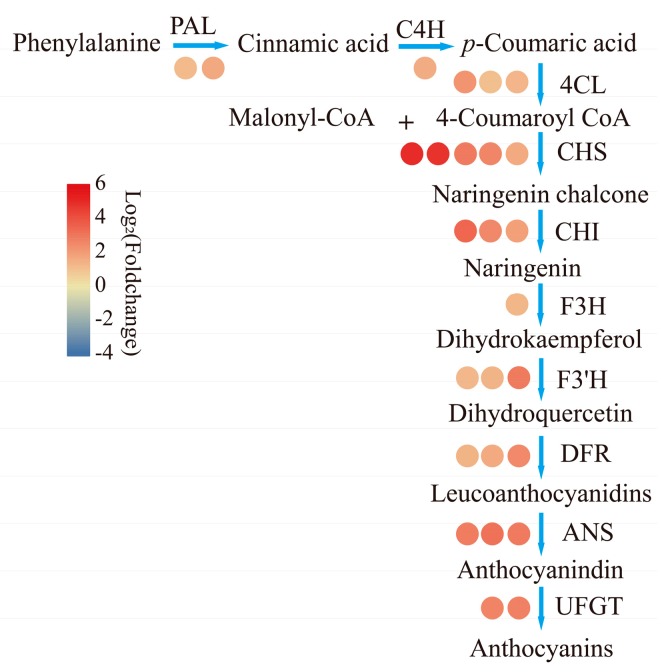
Expression profile of anthocyanin related structural genes. The number of circles represents the gene number and the colors of circle represents the value of log_2_(Foldchange). PAL, phenylalanine ammonium lyase, Cluster-11055.80692, Cluster-11055.82174; C4H, cinnamic acid 4-hydroxylase, Cluster-11055.80529; 4CL, 4-coumaroyl CoA ligase, Cluster-11055.83506, Cluster-11055.81897, Cluster-11055.111791; CHS, chalcone synthase, Cluster-11055.103287, Cluster-11055.148622, Cluster-11055.62221, Cluster-11055.71506, Cluster-11055.81810; CHI, chalcone isomerase, Cluster-11055.10763, Cluster-11055.145351, Cluster-11055.79888; F3H, flavanone 3-hydroxylasel, Cluster-11055.81147; F3’H, flavanone 3′-hydroxylase, Cluster-11055.77085, Cluster-11055.79806, Cluster-11055.82169; DFR, dihydroflavonol reductase, Cluster-11055.59172, Cluster-11055.59175, Cluster-11055.146588; ANS, anthocyanidin synthase, Cluster-11055.79464, Cluster-11055.79498, Cluster-11055.79715; UFGT, UDP-glucose: flavonoid 3-O-glucosyltransferase, Cluster-11055.59183, Cluster-11055.59184.

**Figure 4 ijms-21-01245-f004:**
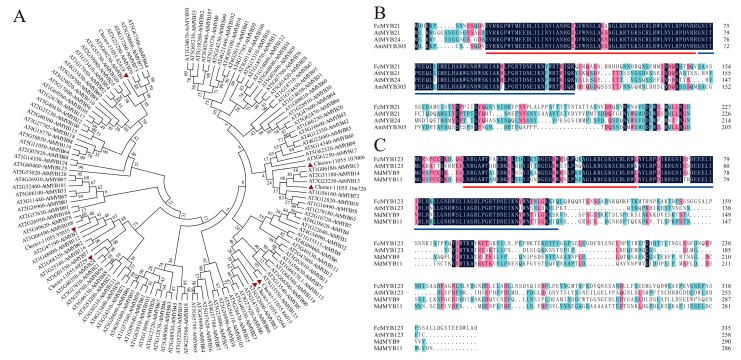
Phylogenetic analysis of fig and *Arabidopsis* R2R3-MYB proteins and multiple sequences alignment of FcMYB21 and FcMYB123. (**A**) Phylogenetic analysis of fig and *Arabidopsis* R2R3-MYB proteins. The red triangles represent the R2R3-MYB proteins of fig. (**B**) Multiple sequences alignment of FcMYB21. (**C**) Multiple sequences alignment of FcMYB123. Identical residues are shown in black, conserved residues in pink (≥ 75%) and similar residues (≥ 50%) in blue in Figure (**B**,**C**). The red and blue lines represent the R2 and R3 motif in Figure (**B**,**C**), respectively. AtMYB21 (*Arabidopsis thaliana*, AT3G27810.1), AtMYB24 (*Arabidopsis thaliana*, AT5G40350.1), AmMYB305 (*Antirrhinum majus*, P81391.1); AtMYB123 (*Arabidopsis thaliana*, AT5G35550.1), MdMYB9 (*Malus* × *domestica*, DQ267900); MdMYB11 (*Malus* × *domestica*, DQ074463).

**Figure 5 ijms-21-01245-f005:**
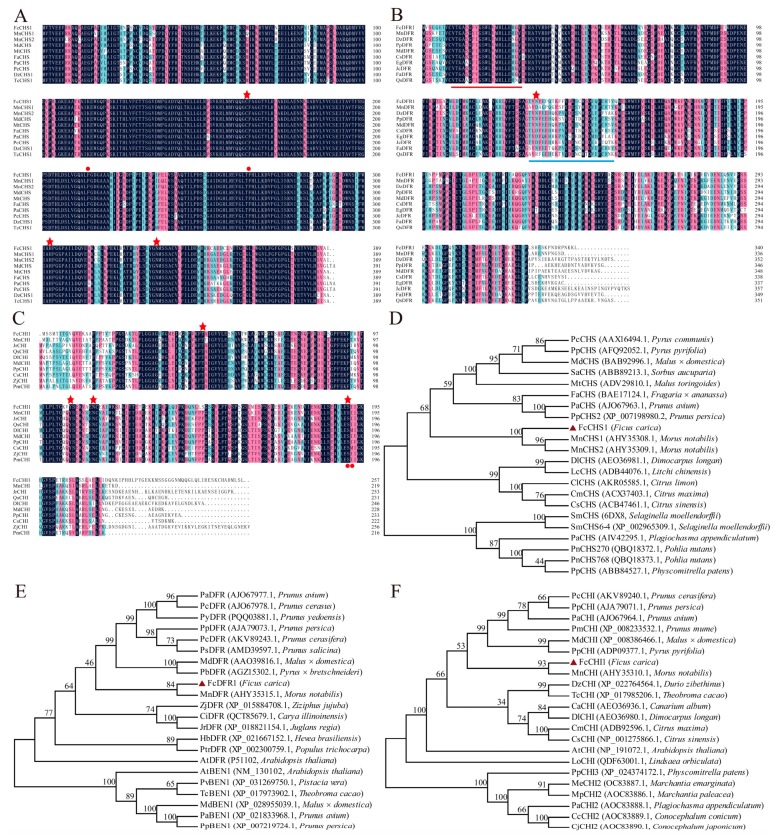
Multiple sequences alignment and phylogenetic analysis of FcCHS1, FcDFR1, and FcCHI1. (**A**) Multiple sequences alignment of FcCHS1. (**B**) Multiple sequences alignment of FcDFR1. The red and blue lines represent NADP binding region and substrate binding region, respectively. The red asterisk represents the key residue in the determination of the substrate specificity of FcDFR1. (**C**) Multiple sequences alignment of FcCHI1. (**D**) Phylogenetic analysis of FcCHS1. (**E**) Phylogenetic analysis of FcDFR1. (**F**) Phylogenetic analysis of FcCHI1. The red asterisks represent the conserved active site and the red dots mean catalytic residues in Figure (**A,C**). Identical residues are shown in black, conserved residues in pink (≥ 75%) and similar residues (≥ 50%) in blue in Figure (**A**–**C**). FcCHS1, FcCHI1 and FcDFR1 protein of fig were marked with red triangles in Figure (**D**–**F**), respectively. CHSs of mosses and Lycophyte, BEN1 proteins and mosses CHIs were used as outgroups in phylogenetic analysis of FcCHS1, FcDFR1 and FcCHS1 in Figure (**D**–**F**), respectively. CHS, chalcone synthase; CHI, chalcone isomerase; DFR, dihydroflavonol reductase, BEN1, BRI1-5 ENHANCED 1. MnCHS1 (*Morus notabilis*, AHY35308.1), MnCHS2 (*Morus notabilis*, AHY35309.1), MdCHS (*Malus* × *domestica*, BAB92996.1), MtCHS (*Malus toringoides*, DV29810.1), FaCHS (*Fragaria* × *ananassa*, BAE17124.1), PaCHS (*Prunus avium*, AJO67963.1), PcCHS (*Pyrus communis*, AAX16494.1), DzCHS1 (*Durio zibethinus*, XP_022721946.1), TcCHS1 (*Theobroma cacao*, XP_007034442.1); MnDFR (*Morus notabilis*, AHY35315.1), DzDFR (*Durio zibethinus*, XP_022769017.1), PpDFR (*Prunus persica*, AJA79073.1), MdDFR (*Malus* × *domestica*, AAO39816.1), CsDFR (*Citrus sinensis*, NP_001275860.1), EgDFR (*Eucalyptus grandis*, XP_010060970.1), JcDFR (*Jatropha curcas*, XP_012071899.1), FaDFR (*Fragaria* × *ananassa*, AHL46451.1), QsDFR (*Quercus suber*, XP_023872028.1); MnCHI (*Morus notabilis*, AHY35310.1), JrCHI (*Juglans regia*, XP_018828033.1), QsCHI (*Quercus suber*, XP_023904011.1), DlCHI (*Dimocarpus longan*, AEO36980.1), MdCHI (*Malus* × *domestica*, XP_008386466.1), PpCHI (*Prunus persica*, XP_007218371.1), CsCHI (*Citrus sinensis*, NP_001275866.1), ZjCHI (*Ziziphus jujube*, XP_015877049.1), PmCHI (*Prunus mume*, XP_008233532.1). The gene IDs and species are in the brackets of phylogenetic analysis.

**Figure 6 ijms-21-01245-f006:**
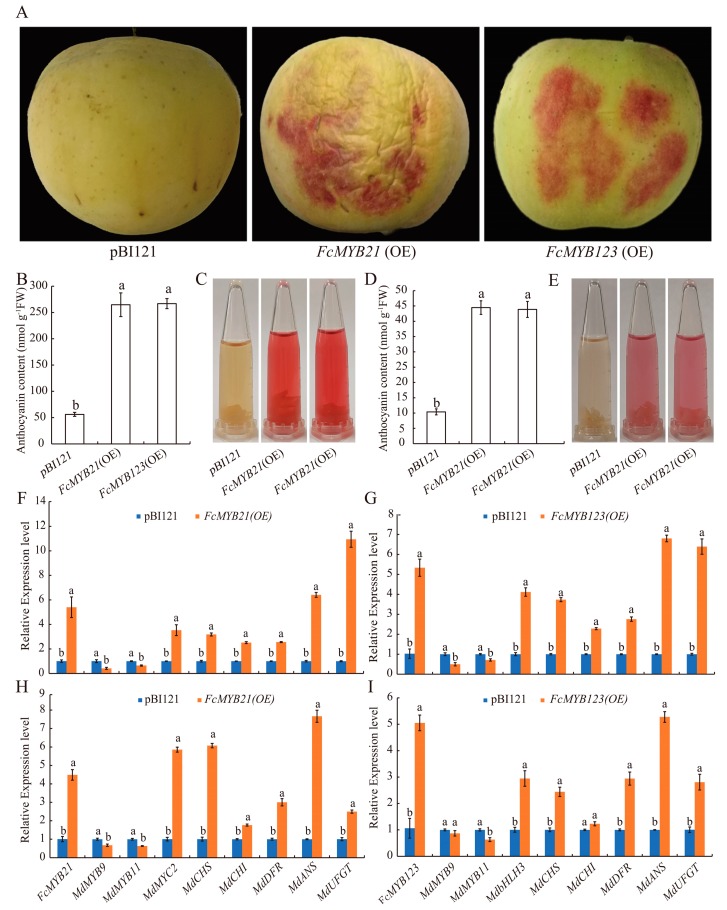
Overexpression of *FcMYB21* and *FcMYB123* in ‘Fuji’ apple peels and calli. (**A**) Overexpressed *FcMYB21* and *FcMYB123* in apple peels. (**B**) The anthocyanin contents in the wild type and transgenic apple peels. (**C**) The colors of anthocyanin extraction in the wild type and transgenic apple peels. (**D**) The anthocyanin contents in the wild type and transgenic apple calli. (**E**) The colors of anthocyanin extraction in the wild type and transgenic apple calli. (**F**) Expression profile of transient overexpression of *FcMYB21* in apple peels. (**G**) Expression profile of transient overexpression of *FcMYB123* in apple peels. (**H**) Expression profile of transient overexpression of *FcMYB21* in apple calli. (**I**) Expression profile of transient overexpression of *FcMYB123* in apple calli. *FcMYB21* (OE) and *FcMYB123* (OE) represent the plasmids with target genes *FcMYB21* and *FcMYB123* in Figure (**A**–**F**), respectively. The empty vector, pBI121 was used as control. The different letters above the same gene represent significant differences (*p* < 0.05).

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
