# Peer review of "Transcriptomic Analysis of Ficus carica Peels with a Focus on the Key Genes for Anthocyanin Biosynthesis"

_ijms, 2020, doi:10.3390/ijms21041245_

Round 1

Reviewer 1 Report

In this article, the authors addressed the molecular process of coloration of fig fruits by means of transcriptome analysis using RNA-seq. Also, the function of two R2R3 MYB transcription factors was evaluated using apple peel and calli. The authors well investigated the previous related works on transcriptome analyses and the fruits coloration in various species and reflected this knowledge in the manuscript. However, it was a little disappointing that this work seemed not quite a novel when considering the previous similar works. The discussion part was too long and sometimes not convincing concerning the results provided in this study, and thus this point should be primarily revised. In addition, many concerns as a scientific publication were found. Here, the major and minor concerns are listed.

Major concerns:

1. The selection of data fo the main and supplement figure/table should be considered again. There too many discussions were made referring supplemental materials. This makes it very difficult for readers to follow the statements of the authors. Here I listed some ideas for improvement.
1.1. Fig 4 qPCR validation should be moved in the supplement since the authors mentioned this data very little. Subsequently, section 2.7 may be united with other sections. Instead, the results for gene expression of the phytohormone-related genes (contents of table S7 and related data) are more relevant to be in the main text. It is much better if it was shown graphically.
1.2. The phylogenetic trees and some alignment data (Fig. S3, S4) can be the main figures, as the authors discussed many things using these relationships.

2. In the analyses of specific categories of genes (i.e., anthocyanin biosynthesis, phytohormone), the authors should include not only DEGs but also non-DEGs, which are annotated as those categories. In the case of phytohormones, for example, there must be a lot of signal factors in each hormone signaling, and only some fraction were differentially expressed. Without the information on how many genes involved in a certain hormone signaling exist in the transcriptome of the plant, you cannot discuss how much impact the detected 34 DEGs have. For some signals, only a few genes fluctuated, and I don't think it is enough discussable for the hormone involvement. The same points may be considered for the anthocyanin biosynthesis, although it seemed not so much problematic than that of the hormone signaling.

3. The exogenous expression using apple are interesting experiments. However, I would like to encourage the authors to examine the expression changes in endogenous MdMYB9 and MdMYB11 in the OE apple peel and calli. It will make clear whether FcMYBs directly regulate Md anthocyanin biosynthesis genes or not. Indirect activation of a gene regulatory module via endogenous counterparts is often a problem for overexpression experiments. Otherwise, the author should discuss such a possibility of the endogenous gene effect.

4. I was a bit worried if the DEG analysis has been done correctly. The Materials and Methods section for the analysis (line 563-569) gave me the impression that the authors used FPKM as input for DEG analysis by DEseq, which would be incorrect. If unnormalized values were inputted correctly, there is no problem. Otherwise, the authors should reanalyze the data.
See DEseq2 manual in detail: http://bioconductor.org/packages/devel/bioc/vignettes/DESeq2/inst/doc/DESeq2.html#why-un-normalized-counts
By the way, reference #106 (citation for DEseq) is the article for another similar program DE"G"seq. Which did you use? (the concern pointed out above may be the case in either of programs.)

5. Overall, the discussion part is too much in terms of both the amount and the extent. Some redundant sentences with the introduction and the results section should be omitted from discussion. More importantly, the authors should focus on the discussion that is well associated and supported by the data provided in this study. Discussions are occasionally too speculative when considering the results and thus less meaningful. Unfortunately, the data from the experimental design of the present study can not provide any direct evidence about the molecular function of genes, so some discussion part should be much toned down.Some specific concerns in the discussion part are listed below.
5.1 line 304-318: The comparison of blast best hit results with previous studies is less meaningful, and this part may be omitted. Because NR (and other) database is being updated, the present situation is likely different from the previous studies unless you intendedly used the same version of database and software.
5.2. line 319-325: Because the total gene number is different, comparisons of just a number of DEGs between species are not meaningful. Also, there may be many differences in not only the coloration process but other associated developmental processes, so it is natural that different species show the different sizes of DEGs. Thus, it is unclear what the authors try to say by these comparisons. (this comment may be related with minor comment #12)
5.3 The authors discussed molecular function using sequence similarity in many parts, but the criteria of the similarity are ambiguous. Maybe it is justified to suggest the same function with quite a high similarity, but with moderate similarity (e.g., 50% at line 412, etc.). Rather, a phylogenetic relationship (orthology) is better supportive information, though it is still not direct.

6. I was really confused by the term "pericarp" (title, line 77, 84, 85, 286, 326, 368, 455, 523) because this particular term indicates the outer tissue of the fruits derived from the ovary. However, from the context, it was supposed to indicate the outer epidermis of the receptacle. In fig, "pericarp" should indicate ovary skin, which is inside of the syconium. As you know, fig's syconium has very peculiar morphology, so the authors have to be careful for readers to understand the material clearly. Just "peel" or "skin" is also insufficient to point the actual part without defining in advance. At least the author should not use "pericarp," and I recommend defining the term once in the text, like, peel (outer epidermis of syconium or receptacle).

minor points:

1. line 15 & 213: The term "Bioinformatic" analysis in these contexts seemed not very appropriate and seemed confusing. Just using "sequence analysis and molecular characteristics" like line 421 is better.
2. line 37 "higher plant": this term is not suitable in the modern scientific view. It should be replaced by the appropriate taxonomical term, such as vascular plants.
3. line 104-112, "...genes were annotated in ... database": this statement sounds strange. "a gene having blast (or another search) hit" is not equal to "being annotated." Precisely, these genes may have been annotated according to the blast (best) hit results.
4. Figure 2B, Color assignment should be based on the log10-transformed p-value, and the range of color should be adjusted to the actual data range. Almost no difference is recognized in the present color code.
5. line 191: What "the seven necessary members" are. Please specify them.
6. In section 2.8 and Supplemental Table 9, the molecular characteristics of the proteins whose function for anthocyanin biosynthesis has been proven (e.g., orthologs of Arabidopsis) should also be mentioned and compared. Just referring to fig proteins gave us no information about its function.
7. line 228, "homology was 91.54%" & line 239 "high homology": the term "homology" should be replaced by "similarity" or "identity". Homology cannot be presented by the extent (such as % and high/low).
8. Figure 5. The order of the figure is tricky and hard to understand at a glance. Please assign caption (A, B, C, and so forth) from the left-top to right-bottom naturally. In the main text, it is less problematic if the figures are mentioned in a different order from the figure.
9. line 297: 186,084 unigenes seems quite a large number compared to the gene number of model plants in which genomic sequences are open, though I understand that de novo assembly by trinity tends to output such a large number of genes. I wondered there was a recent genome duplication event in fig evolution. If so, it explains the large unigene number, and this information is worth to be written in the text.
10. line 298-300: the stats of contig length is an index to compare the assembly quality, but not a better index. If you can use the assembly of previous studies, the conserved gene completeness calculated by BUSCO software will serve a more informative comparison, for example. This is just a recommendation.
11. line 316-317, "However, using the genomic database... The value confirms": The background of these statements is unclear. Was this result acquired by a blast against only M. notabilis proteome or transcriptome? If so, it is not comparable to the blast result against NR in which thousands of species are included. In addition, blast best hit species are often not the closest species in many proteins. The following sentence is also unnecessary unless the relationship between them was controversial. I think their close relationship is already proven in some way, as they are in the same family.
12. line 325, "the number of DEGs are sufficient...": What is the sufficient number of DEGs? This statement seemed to represent the authors' preconception as if the detection of a small number of DEGs does not reflect the actual biological phenomenon. A large number of DEGs is not always better. Though I understand some downstream analyses such as GO term enrichment analysis requires some amount of a gene set, this statement seemed not very scientific.
13. line 328-330, "the most genes related ...": This was not able to be concluded from the data, because the authors seemingly did not show all genes related to these hormone signaling. This comment is related to major concern #2. The author should consider hormonal involvement after examining the data, including that of non-DEG genes.
14. line 359 "ETR ...": this sentence seemed not appropriate for ETRs. Superficially, the absence of ETRs activates ET signaling via inhibiting the degradation of EIN3, but they are receptor, so I think it is inappropriate to classify ETRs as negative factors of ET signaling. Saying EBF so is fine.
15. line 376-377, "implying that there may be more than ...": The meaning of this statement seems needless to say since the authors actually found two PALs and three 4CLs even in DEGs, and they may be found more in non-DEGs.
16. line 453-454, "This situation means...": This sentence does not make sense. First, the tree is unrooted and has adjusted branch length, so people cannot tell the ancestry nor conservation from the tree. Second, in principle, all homologous genes equally related to their common ancestor through evolution because it is "common" to these proteins. I recommend including some outgroup genes to the phylogenetic analysis to prove whether FcDFR1 is truely the ortholog of known DFR1genes.
17. Many company locations shown in Materials and Methods parts were incorrect. e.g., TOYOBO (line 584) and Takara (line 587) are in Japan, and GENEWIZ (line 589) and ThermoFisher Scientific (line 592) are in the US, for the head offices. There may be other cases. Please clarify the company information again.
18. line 557-558, "Then selected ...": When considering the software function, the sentence must be revised like "The transcripts were clustered by hierarchical cluster analysis [103] and then the longest transcript sequence was selected from each cluster as unigene."
19. line 559-562: As mentioned in minor point 3, usage of the term "annotate" should be revised. I suppose that the authors used BLASTs (Blastn, Blastp, or Blastx ??) and hmm search to annotate genes. So the authors should describe the usage of these software (e.g., threshold e-values, etc.).
20. line 579: The citation for MEGA is lacking.
21. Reference #105 is missing in the main text.
22. The gene names after the orthology and the names after assembly (e.g., FcMYB21 and Cluster-11055.69229) are mixed up in the manuscript, and that makes it hard to understand. Please unify the usage of gene names.
23. The RNA-seq reads should be deposited to an appropriate database (NCBI/EBI/DDBJ), and the accession numbers should have appeared in accordance with the journals' policy.

Author Response

Response to Reviewer 1 Comments

Point 1: Moderate English changes are required.

 Response 1: Thanks for your kindly suggestion. The language of this paper had been comprehensively modified by a native English speaker. Hopefully it meets the requirements for publication in this journal.

Major concerns

Point 1: The selection of data for the main and supplement figure/table should be considered again. There too many discussions were made referring supplemental materials. This makes it very difficult for readers to follow the statements of the authors. Here I listed some ideas for improvement.

Response 1: Thanks for your constructive suggestion. We have been carefully considered your suggestion and adjusted the supplementary materials. The results are as follows:

Point 1.1: Fig 4 qPCR validation should be moved in the supplement since the authors mentioned this data very little. Subsequently, section 2.7 may be united with other sections. Instead, the results for gene expression of the phytohormone-related genes (contents of table S7 and related data) are more relevant to be in the main text. It is much better if it was shown graphically.

Response 1.1: We have transferred the original Figure 4 from the main text to the supplemental materials (Figure S3).

We have also united section 2.7 with 2.3 (Analysis of Differentially Expressed Genes) in Page 5.

The results for gene expression of the phytohormone-related has been deleted from the main text. The reasons were listed in Major concerns 2.

Point 1.2: The phylogenetic trees and some alignment data (Fig. S3, S4) can be the main figures, as the authors discussed many things using these relationships.

Response 1.2: Thanks for your suggestion. We have transferred the original Figure S3 and Figure S4 to the main text. Please see Figure 4 and Figure 5 in the new manuscript (Page 5 and Page 7).

Point 2: In the analyses of specific categories of genes (i.e., anthocyanin biosynthesis, phytohormone), the authors should include not only DEGs but also non-DEGs, which are annotated as those categories. In the case of phytohormones, for example, there must be a lot of signal factors in each hormone signaling, and only some fraction were differentially expressed. Without the information on how many genes involved in a certain hormone signaling exist in the transcriptome of the plant, you cannot discuss how much impact the detected 34 DEGs have. For some signals, only a few genes fluctuated, and I don't think it is enough discussable for the hormone involvement. The same points may be considered for the anthocyanin biosynthesis, although it seemed not so much problematic than that of the hormone signaling.

Response 2: We agree with that "Without the information on how many genes involved in a certain hormone signaling exist in the transcriptome of the plant, we cannot discuss how much impact the detected 34 DEGs have". We analysed all the unigenes that involved in hormone signalling and calculated the percentage of DEGs to all unigenes related to phytohormones. The proportion was too low (2-5%) to draw a clear conclusion of the effect of hormones. Therefore, we have deleted the hormone related introduction, results and discussion from the main text.

We also calculated the percentage of DEGs to all unigenes that related to anthocyanin and found that a large portion of unigenes related to anthocyanin were differently expressed between the two stages. We have put this result in the new manuscript (Line 127-128). "There were 152 anthocyanin related unigenes. Among them identified 73 DEGs, there are 26 DEGs left after excluding 47 short fragment of CHSs."

Point 3: The exogenous expression using apple are interesting experiments. However, I would like to encourage the authors to examine the expression changes in endogenous MdMYB9 and MdMYB11 in the OE apple peel and calli. It will make clear whether FcMYBs directly regulate Md anthocyanin biosynthesis genes or not. Indirect activation of a gene regulatory module via endogenous counterparts is often a problem for overexpression experiments. Otherwise, the author should discuss such a possibility of the endogenous gene effect.

Response 3: Thanks for your valuable suggestion. We examined the expressions of endogenous MdMYB9 and MdMYB11 together with the biosynthetic genes when we did the experiments. But we did not put the results in the original manuscript. Now, we have added these results to Figure 6. We also added some results and discussion correspondingly in the Result section 2.8 and Discussion section 3.3.

Result 2.8 (Line 261-266): "The expression levels of MdMYB9 and MdMYB11 in FcMYB21- and FcMYB123-overexpressed apple peels and calli decreased or did not change in comparison with wild type. Differently, the transcription level of MdMYC2 and MdbHLH3 increased sharply in FcMYB21- and FcMYB123-overexpressed apple peels and calli. These results indicated that the effect of FcMYB21 and FcMYB123 may be independent of endogenous MdMYB9 and MdMYB11, but related to bHLH partners."

Discussion 3.3 (Line 422-430): "The downregulation of MdMYB9 and MdMYB11 in transgenic apple peels and calli was similar to the result of overexpression of LcMYB1 in tobacco, which induced anthocyanin accumulation in flower but repressed the expression of NtAn2, an anthocyanin related MYB gene, in flower [72]. The results indicate that the effect of FcMYB21 and FcMYB123 on anthocyanin biosynthesis may be independent of endogenous MdMYB9 and MdMYB11. The bHLH partners, MdMYC2 and MdbHLH3 was significantly up-regulated in overexpression of FcMYB21 and FcMYB123, respectively. The results was similar with MrMYB1. Overexpression of MrMYB1 in Arabidopsis induced anthocyanin accumulation in leaves and AtTT8 was significant upregulated [73]".

Reference 72:

Lai,B.; Li, X.J.; Hu, B.; Qin, Y.H.; Huang, X. M.; Wang, H.C.; Hu, G.B. LcMYB1 is a key determinant of differential anthocyanin accumulation among genotypes, tissues, developmental phases and ABA and light stimuli in Litchi chinensis. PLoS One. 2014, 21. 1-12

Reference 73:

Huang, Y.J.; Song, S.; Andrew C.A.; Liu, X.F.; Yin, X.R.; Xu, C.J.; Chen, K.S. Differential activation of anthocyanin biosynthesis in Arabidopsis and tobacco over-expressing an R2R3 MYB from Chinese bayberry. Plant Cell Tiss Org. 2013, 113, 491-499.

Point 4: I was a bit worried if the DEG analysis has been done correctly. The Materials and Methods section for the analysis (line 563-569) gave me the impression that the authors used FPKM as input for DEG analysis by DEseq, which would be incorrect. If unnormalized values were inputted correctly, there is no problem. Otherwise, the authors should reanalyze the data.
See DEseq2 manual in detail: http://bioconductor.org/packages/devel/bioc/vignettes/DESeq2/inst/doc/DESeq2.html#why-un-normalized-counts
By the way, reference #106 (citation for DEseq) is the article for another similar program DE"G"seq. Which did you use? (the concern pointed out above may be the case in either of programs.)

Response 4: We are sorry for our miswriting. We have checked the information with Novogene company and ensured that we used readcount values and DEseq2 in DEGs analysis. We have revised the methods in the Materials and Methods section 4.4 and corrected the corresponding reference [82].

Materials and Methods section:

(Line 492): "The input data for DEGs analysis was the readcount values."

(Line 495): "Differential expression analysis of two development stages (contains three biological replicates) was performed using the DESeq2 [82]"

Reference 82:

Love, M.I.; Huber, W.; Anders, S. Moderated estimation of fold change and dispersion for RNA-seq data with DEseq2. Geno Biol. 2014, 15, 550-571.

Point 5: Overall, the discussion part is too much in terms of both the amount and the extent. Some redundant sentences with the introduction and the results section should be omitted from discussion. More importantly, the authors should focus on the discussion that is well associated and supported by the data provided in this study. Discussions are occasionally too speculative when considering the results and thus less meaningful. Unfortunately, the data from the experimental design of the present study cannot provide any direct evidence about the molecular function of genes, so some discussion part should be much toned down. Some specific concerns in the discussion part are listed below.

Response 5: We have shortened the Discussion according to your following suggestion.

Point 5.1: line 304-318: The comparison of blast best hit results with previous studies is less meaningful, and this part may be omitted. Because NR (and other) database is being updated, the present situation is likely different from the previous studies unless you intendedly used the same version of database and software.

Response 5.1: We have deleted this part according to your suggestion.

Point 5.2 line 319-325: Because the total gene number is different, comparisons of just a number of DEGs between species are not meaningful. Also, there may be many differences in not only the coloration process but other associated developmental processes, so it is natural that different species show the different sizes of DEGs. Thus, it is unclear what the authors try to say by these comparisons. (this comment may be related with minor comment #12)

Response 5.2: We have deleted this part from the Discussion section.

Point 5.3 The authors discussed molecular function using sequence similarity in many parts, but the criteria of the similarity are ambiguous. Maybe it is justified to suggest the same function with quite a high similarity, but with moderate similarity (e.g., 50% at line 412, etc.). Rather, a phylogenetic relationship (orthology) is better supportive information, though it is still not direct.

Response 5.3:Thank you for your suggestion. The criteria of similarity are ambiguous, which is related to the conservation of genes. It is hard to point out a specific standard without specifying a gene family. To avoid misleading readers, we have largely decreased the use of sequence similarity in the Result and Discussion section.

The specific revisions were showed as follows:

(1) We deleted the sentence "And then, its homology was 91.54% with MiCHS1 of Mangifera indica, and 90.54% of PeCHS1 of Populus euphratica. These results suggest that FcCHS1 is a very conserved gene among plant species".

(2) "In this study, the amino acid sequence of Cluster-11055.81897 (4CL) showed a close relationship with At4CL3 (69.03%) and Mn4CL3 (78.66%), and therefore was considered as one of the candidate genes for pigmentation in fig" was changed to "In this study, the amino acid sequence of Cluster-11055.81897 (4CL) showed a close relationship with Mn4CL3 , and therefore was considered as one of the candidate genes for pigmentation in fig" (Line 311-312).

(3) "Fig F3H (Cluster-11055.81147) and F3'H (Cluster-11055.79806) had high similarity with AtTT6 and AtTT7, respectively, suggesting their possible roles in anthocyanin biosynthesis" was changed to "The blast best hit Arabidopsis thaliana genes of F3H (Cluster-11055.81147) and F3'H (Cluster-11055.79806) were AtTT6 and AtTT7, respectively, suggesting their possible roles in anthocyanin biosynthesis" (Line 330-332).

(4) "The predicted polypeptide sequence of Cluster-11055.59183 mapped about 50% identity to MaUFGT2, a putative major gene in regulation anthocyanin synthesis in Morus alba, indicating that Cluster-11055.59183 may play the same role in fig." was changed to "Although both composed a close relationship with MaUFGT2, a putative major gene in regulation anthocyanin synthesis in Morus alba, the predicted polypeptide sequence of Cluster-11055.59183 mapped a higher similarity to MaUFGT2, indicating that Cluster-11055.59183 may play the similar role in fig" (Line 341-344).

(5) "The results suggest that FcCHS1 and FcCHI1 encode typical CHS and CHI protein, respectively. FcCHS1 shared 94.86% similarity with MnCHS1 and MnCHS2. Among the three annotated CHI genes, the coding sequence of Cluster-11055.10763 harbored the highest matching rate with MnCHI (64.20%). MnCHS1, MnCHS2, and MnCHI had been proved to be involved in anthocyanin accumulation in flowers of the overexpressing tobacco lines" and "Similar result has been reported of GbDFR1 which exhibits a high identity (70%-80%) with the DFRs from other plants" was changed to "The same conserved amino acid residues of FcCHS1, FcCHI1 and FcDFR1 as the corresponding genes of other vascular plants indicate that they are truly the ortholog of known proteins in the three families. Moreover, FcCHS1, FcCHI1 and FcDFR1 shared high similarity and had a close relationship with counterparts of Morus notabilis. MnCHS1, MnCHS2, MnCHI and MnDFR1 had been proved to be involved in anthocyanin accumulation in flowers of the overexpressing tobacco lines. The three cloned genes may perform the same function in fig coloration" (Line 374-379).

Point 6: I was really confused by the term "pericarp" (title, line 77, 84, 85, 286, 326, 368, 455, 523) because this particular term indicates the outer tissue of the fruits derived from the ovary. However, from the context, it was supposed to indicate the outer epidermis of the receptacle. In fig, "pericarp" should indicate ovary skin, which is inside of the syconium. As you know, fig's syconium has very peculiar morphology, so the authors have to be careful for readers to understand the material clearly. Just "peel" or "skin" is also insufficient to point the actual part without defining in advance. At least the author should not use "pericarp," and I recommend defining the term once in the text, like, peel (outer epidermis of syconium or receptacle).

Response 6: To avoid misleading readers, we have changed "pericarp" and "skin" to "peel" in the whole manuscript (title, Line 10, 28, 29, 32, 66, 71, 73, 74, 75, 186, 249, 250, 252, 253, 262, 264, 268, 269, 270, 272, 273, 275, 280, 297, 300, 302, 350, 370, 380, 391, 416, 418, 420, 423, 448, 451, 452, 462, 463, 470, 512, 522, 540, 541).

In the Materials and Methods (4.1), we have added clear definition of "peel": Fig fruit peels refer to the tissues outside the receptacle (Line 453).

minor points:

Point 1: line 15 & 213: The term "Bioinformatic" analysis in these contexts seemed not very appropriate and seemed confusing. Just using "sequence analysis and molecular characteristics" like line 421 is better.

Response 1: We have replaced the term "Bioinformatic" by "physicochemical properties and structure characteristics" (Line 15, 184 ).

Point 2: line 37 "higher plant": this term is not suitable in the modern scientific view. It should be replaced by the appropriate taxonomical term, such as vascular plants.

Response 2: Thank you for you constructive suggestions. The term "higher plant" was replaced by "vascular plants" (Line 38).

Doyle, J.A. Phylogeny of vascular plants. Annu Rev Ecol Syst. 1998, 29, 567-599.

Point 3: line 104-112, "...genes were annotated in ... database": this statement sounds strange. "a gene having blast (or another search) hit" is not equal to "being annotated." Precisely, these genes may have been annotated according to the blast (best) hit results.

Response 3: To make the expression more accurate, we have changed "154,920 unigenes were annotated in at least one databases" to "154,920 unigenes were matched homologous sequences in at least one databases " (Line 94).

We changed "138,194 (74.27%) sequences were annotated with reference to..." to "74.27% (138,194) sequences were mapped to known proteins in the..." (Line 95).

We changed "(14.99%) were annotated in all databases..." to "the mapping ratio were 14.99% (27,893) in all databases" (Line 96).

We changed "Almost 68.20% of the annotated sequences bore E-value < 10−45..." to "68.20% of the mapped genes in Nr database showed E-value < 10−45" (Line 98).

Point 4: Figure 2B, Color assignment should be based on the log10-transformed p-value, and the range of color should be adjusted to the actual data range. Almost no difference is recognized in the present color code.

Response 4: We have revised Figure 2B according to your comment. The color assignment was conducted based on the log10-transformed p-value, and the range of color was adjusted to the actual data range.

Point 5: line 191: What "the seven necessary members" are. Please specify them.

Response 5: We are sorry for confusing you. We have deleted the sentence (Major concerns 2).

Point 6: In section 2.8 and Supplemental Table 9, the molecular characteristics of the proteins whose function for anthocyanin biosynthesis has been proven (e.g., orthologs of Arabidopsis) should also be mentioned and compared. Just referring to fig proteins gave us no information about its function.

Response 6: Thank you for your useful suggestion. We carried out the physicochemical properties and secondary structure of the proteins whose function have been proven in anthocyanin biosynthesis in Table S8. The following related analysis were listed in the Result 2.6, 2.7 and Discussion 3.2, 3.3 section in the new manuscript.

Result 2.6: "Comparing them with their orthologs whose function for anthocyanin biosynthesis have been proved, we found they all are acidic amino acid and hydrophilic protein. Their proportion of secondary structure are extremely similar and the variation is less than 3.0%" (Line 194-196).

Result 2.7: "FcMYB21 and FcMYB123 composed the similar composition of secondary structure with those MYBs that have been proved regulating anthocyanin accumulation" (Line 247-248).

Discussion 3.2: "The similar molecular characteristics and proportion of secondary structure of FcCHS1, FcCHI1, FcDFR1 with their orthologs proteins whose function have been proved in anthocyanin accumulation imply their role in fig coloring (Table S8)" (Line 352-354).

Discussion 3.3: (1) " Here, Cluster-11055. 86015 harbors high similarity in secondary structure with MdMYB9... " (Line 400-401).

Discussion 3.3: (2) "...with AmMYB305 and composed similar secondary structure with it" (Line 405-406).

Point 7: line 228, "homology was 91.54%" & line 239 "high homology": the term "homology" should be replaced by "similarity" or "identity". Homology cannot be presented by the extent (such as % and high/low).

Response 7: We have deleted "homology was 91.54%" (Major concerns 2).

We revised "high homology" to "high identity" (Line 210).

Point 8: Figure 5. The order of the figure is tricky and hard to understand at a glance. Please assign caption (A, B, C, and so forth) from the left-top to right-bottom naturally. In the main text, it is less problematic if the figures are mentioned in a different order from the figure.

Response 8: We have assigned the caption of this Figure from the left-top to the right-bottom to make it clearer according to your suggestion (Figure 6 in the new manuscript).

Point 9: line 297: 186,084 unigenes seems quite a large number compared to the gene number of model plants in which genomic sequences are open, though I understand that de novo assembly by trinity tends to output such a large number of genes. I wondered there was a recent genome duplication event in fig evolution. If so, it explains the large unigene number, and this information is worth to be written in the text.

Response 9: The reasons for a large number of unigenes in this study were listed as follows.

First of all, compared with apple and pear, whose size stop changing during the coloring process, fig fruits will continue expanding during the coloration period. Therefore, anthocyanin accumulation of fig fruits were accompanied with many other biological processes, resulting lots of unigenes.

Secondly, the fig fruits selected for transcriptomic analysis showed great difference in color.

Similar result has been reported previously. For example, 179,951 unigenes were obtained from leaves of ‘Anji Baicha’ (Camellia sinensis) in different development stages.

The whole genome database of fig has not been published and the analysis of fig whole genome (Horaishi) did not cover recent genome duplication event in fig evolution. Therefore, we were unable to provide this information in this paper.

Reference:

Li, C.F.; Xu, Y.X.; Ma, J.Q.; Jin, J.Q.; Chen, L. Biochemical and transcriptomic analyses reveal different metabolite biosynthesis profiles among three color and developmental stages in ‘Anji Baicha’ (Camellia sinensis). BMC Plant Biol. 2016, 16, 195-212.

Mori, K.; Shirasawa, K.; Nogata, H.; Hirata, C.; Tashiro, K.; Habu, T.; Kim, S.; Himeno, S.; Kuhara, S.; Ikegami, H. Identification of RAN1 orthologue associated with sex determination through whole genome sequencing analysis in fig (Ficus carica L.). Sci Rep. 2017, 7, 41124-41136.

Point 10: line 298-300: the stats of contig length is an index to compare the assembly quality, but not a better index. If you can use the assembly of previous studies, the conserved gene completeness calculated by BUSCO software will serve a more informative comparison, for example. This is just a recommendation.

Response 10: The stats of contig length is now a common index to compare the assembly quality which has been used by many researchers (Litchi chinensis and Ficus carica). So, we used this index. We will try BUSCO software in future research.

Lai, B.; Hu, B.; Qin, Y.H.; Zhao, J.T.; Wang, H.C.; Hu, G.B. (2015). Transcriptomic analysis of Litchi chinensis pericarp during maturation with a focus on chlorophyll degradation and flavonoid biosynthesis. BMC Geno. 2015, 16, 225-243.

Wang, Z.R.; Cui, Y.Y.; Vainstein, A.; Chen, S.W.; Ma, H.Q. Regulation of fig (Ficus carica L.) fruit color: metabolomic and transcriptomic analyses of the flavonoid biosynthetic pathway. Front Plant Sci. 2017, 8, 1990-2005.

Point 11: line 316-317, "However, using the genomic database... The value confirms": The background of these statements is unclear. Was this result acquired by a blast against only M. notabilis proteome or transcriptome? If so, it is not comparable to the blast result against NR in which thousands of species are included. In addition, blast best hit species are often not the closest species in many proteins. The following sentence is also unnecessary unless the relationship between them was controversial. I think It their close relationship is already proven in some way, as they are in the same family.

Response 11: The genomic database used here was proteome of M. notabilis.

We agree with your suggestion and deleted this part from the Discussion.

Point 12: line 325, "the number of DEGs are sufficient...": What is the sufficient number of DEGs? This statement seemed to represent the authors' preconception as if the detection of a small number of DEGs does not reflect the actual biological phenomenon. A large number of DEGs is not always better. Though I understand some downstream analyses such as GO term enrichment analysis requires some amount of a gene set, this statement seemed not very scientific.

Response 12: To avoid misleading readers, we have deleted this sentence.

Point 13: line 328-330, "the most genes related ...": This was not able to be concluded from the data, because the authors seemingly did not show all genes related to these hormone signaling. This comment is related to major concern #2. The author should consider hormonal involvement after examining the data, including that of non-DEG genes.

Response 13: We agree with you that the background genes is a very important factor in considering the effects of hormone.

We have deleted the results and discussion of plant hormone and the reasons were listed in Major concerns 2.

Point 14: line 359 "ETR ...": this sentence seemed not appropriate for ETRs. Superficially, the absence of ETRs activates ET signaling via inhibiting the degradation of EIN3, but they are receptor, so I think it is inappropriate to classify ETRs as negative factors of ET signaling. Saying EBF so is fine.

Response 14: We have deleted the results of plant hormone and the reasons were listed in Major concerns 2.

Point 15: line 376-377, "implying that there may be more than ...": The meaning of this statement seems needless to say since the authors actually found two PALs and three 4CLs even in DEGs, and they may be found more in non-DEGs.

Response 15: Thank you for your reminding. We revised the sentence " implying that there may be more than ..." to "indicating that there are more than one genes in fig PAL and 4CL family" (Line 453).

Point 16: line 453-454, "This situation means...": This sentence does not make sense. First, the tree is unrooted and has adjusted branch length, so people cannot tell the ancestry nor conservation from the tree. Second, in principle, all homologous genes equally related to their common ancestor through evolution because it is "common" to these proteins. I recommend including some outgroup genes to the phylogenetic analysis to prove whether FcDFR1 is truely the ortholog of known DFR1genes.

Response 16: Thank you for your suggestion and detailed explanation. Some new genes have been added to prove FcDFR1 is truely the ortholog of known DFR1genes (Figure 5E) according to your suggestion. Besides, the sentence "This situation means..." has been deleted.

Point 17: Many company locations shown in Materials and Methods parts were incorrect. e.g., TOYOBO (line 584) and Takara (line 587) are in Japan, and GENEWIZ (line 589) and ThermoFisher Scientific (line 592) are in the US, for the head offices. There may be other cases. Please clarify the company information again.

Response 17: We used the location of the distributor before, and now we have changed them to the location of the head office.

We revised "TIANGEN, Beijing, China" as "TIANGEN, Atlanta, US" (Line 472).

We revised "TOYOBO, Shanghai, China" as "TOYOBO, Osaka, Japan" (Line 472).

We revised "TaKaRa, Nanjing, China" as "TaKaRa, Shiga, Japan" (Line 514).

We revised "Transgen Biotech, Nanjing, China" as "Transgen Biotech, Beijing, China" (Line 515).

We revised "GENEWIZ (Yangzhou, China)" as "GENEWIZ (New Jersey, US)" (Line 516).

We revised "Thermo Scientific, Shanghai, China" as "Thermo Scientific, Waltham, US" (Line 519).

We revised "Transgen Biotech, Nanjing, China" as "Transgen Biotech, Beijing, China" (Line 519).

Point 18: line 557-558, "Then selected ...": When considering the software function, the sentence must be revised like "The transcripts were clustered by hierarchical cluster analysis [103] and then the longest transcript sequence was selected from each cluster as unigene."

Response 18: Thank you for your suggestions We have revised the sentence "Then selected ..." to "The transcripts were clustered by hierarchical cluster analysis [78] and then the longest transcript sequence was selected from each cluster as unigene" (Line 484).

Point 19: line 559-562: As mentioned in minor point 3, usage of the term "annotate" should be revised. I suppose that the authors used BLASTs (Blastn, Blastp, or Blastx ??) and hmm search to annotate genes. So the authors :should describe the usage of these software (e.g., threshold e-values, etc.).

Response 19: We have revised the term "annotate" to "map" or "match" in the full text. The detailed information was listed in minor point 3.

In 4.3, we provided the software and thresholds used in the seven databases as follows. "To identify putative functions of fig unigenes, we performed blastx against the Nr database, the NCBI non-redundant nucleotide sequences (Nt), clusters of orthologous groups of proteins (KOG) and Swiss-Prot with E-value < 10−5. The protein family (Pfam) was carried out by HMMER with E-value < 10−2. KEGG was taken by KEGG automatic annotiation server with E-value < 10−10 and GO databases was based on Nr and PFam by Blast2GO v2.5 (E-value < 10−6) [79]" (Line 486-490).

Point 20: line 579: The citation for MEGA is lacking.

Response 20: We have added the reference to MEGA software, reference 83 (Line 507).

Tamura, K.; Peterson, D.; Peterson, N.; Stecher, G.; Nei, M.; Kumar, S. Mega5: molecular evolutionary genetics analysis using maximum likelihood, evolutionary distance, and maximum parsimony methods. Mol Biol Evol. 2011, 28, 2731-2739.

Point 21: Reference #105 is missing in the main text.

Response 21: We have rechecked the whole article and confirmed that reference #105 is on Line 675 in the main text of revised file with track changes.

Point 22: The gene names after the orthology and the names after assembly (e.g., FcMYB21 and Cluster-11055.69229) are mixed up in the manuscript, and that makes it hard to understand. Please unify the usage of gene names.

Response 22: We standardized the description process, using "cluster--" for transcriptome data analysis and utilizing the gene names of cloned genes in later analysis.

Point 23: The RNA-seq reads should be deposited to an appropriate database (NCBI/EBI/DDBJ), and the accession numbers should have appeared in accordance with the journals' policy.

Response 23: We have deposited our RNA-seq reads to NCBI. However, we haven't received the email from NCBI to get the access number yet. The submission number is SUB6889816 and the information will be supplemented once we get the access number.

Reviewer 2 Report

The aim of this study is to analyse the key biosynthetic and regulatory genes of anthocyanin metabolism in fig (Ficus carica) using high-throughput RNA-sequencing. Several upregulated genes were transiently transformed to apple fruits and apple calli to test their roles in anthocyanin accumulation. Such research is valuable for fig researchers, however, as authors state, genes in anthocyanins biosynthesis pathway are conservative, and are very similar in higher plants. In my opinion, manuscript is well prepared, huge research was performed, but manuscript is technical, and novelty of presented data should be highlighted more. English language in general is good, but there are some mistakes in the text.

All New high throughput sequencing (HTS) should be deposited either in the GEO database or in the NCBI’s Sequence Read Archive according to instruction to authors, and that should be stated in the text.

hm2 as an area unit is not very common, I suggest using ha.

Authors evaluated general anthocyanin concentration in extracts, composition of individual anthocyanins would provide additional valuable information.

Also, more data acquisition points during ripening would provide dynamics  of expression of already cloned anthocyanin biosynthesis genes.

Conclusions should be more specific, they just repeat result section. Conclusions should be aimed at presenting new insights and novel knowledge.

In general, manuscript may be published after minor revision.

Author Response

Response to Reviewer 2 Comments

Point 1: English language and style are fine/minor spell check required.

Response 1: Thanks for your kindly suggestion. The language of this paper had been comprehensively modified by a native English speaker. Hopefully it meets the requirements for publication in this journal.

Point 2: All New high throughput sequencing (HTS) should be deposited either in the GEO database or in the NCBI’s Sequence Read Archive according to instruction to authors, and that should be stated in the text.

Response 2: We have deposited our RNA-seq reads to NCBI. However, we haven't received the email from NCBI to get the access number yet. The submission number is SUB6889816 and the information will be supplemented once we get the access number.

Point 3: hm2 as an area unit is not very common, I suggest using ha.

Response 3: We have revised hm2 to ha (Line 62).

Point 4: Authors evaluated general anthocyanin concentration in extracts, composition of individual anthocyanins would provide additional valuable information. Also, more data acquisition points during ripening would provide dynamics of expression of already cloned anthocyanin biosynthesis genes.

Response 4: The composition of individual anthocyanins would provide additional valuable information in fig anthocyanin accumulation. However, our main aim is providing basic molecular bases of fig coloration by RNA-seq. We have screened some relevant biosynthetic genes and transcription factors, providing some bases for further deep research. The composition of individual anthocyanins will be detected in our further deep research on fig anthocyanin regulation.

The time from the yellow stage of figs to the commodity maturity is very short, which can be completed within only three days when the temperature is high. This situation makes it difficult to distinguish the middle process of fig coloring by eyes. Therefore, here, we just choose two stages for experiment.

Point 5: Conclusions should be more specific, they just repeat result section. Conclusions should be aimed at presenting new insights and novel knowledge.

Response 5: Thanks for your suggestion. We have revised the conclusion as follows to make it more specific.

Based on RNA-seq, we cloned three structural genes, FcCHS1, FcCHI1, and FcDFR1, which are candidate genes in anthocyanin biosynthesis. Furthermore, transient expression of FcMYB21 or FcMYB123, two R2R3-MYBs, in apple demonstrated their regulatory roles in anthocyanin accumulation. Although more verification about their function in fig fruits needs be further conducted, our data provide some molecular basis for fig anthocyanin biosynthesis (Line 438-443).

Round 2

Reviewer 1 Report

I think the manuscript has been significantly improved, but I found a few points that have to be reconsidered. After considering these points, the manuscript will be ready to publish.

Related to the response to minor point 9 (Figure 5):
Fig. 5E tree still does not support orthology of the fig gene, because the authors included just additional orthologous genes, but informative "outgroup" genes. From the topology of this unrooted tree, there is still a possibility that the clade of Fc and Mn genes is clustered with the outer sister family. To show the clear orthology of these genes to other known genes, I have commented that the authors need to add appropriate "outgroup genes" to root the tree. The outgroup can be either non-orthologous homolog genes, which are the closest sister of this gene family, or basal plant's orthologs.
For example, outgroup for DFR can be Arabidopsis BEN1(AT2G45400.1) and some orthologs of it.
The trees of CHS and CHI are also the same case. However, they are likely quite old genes, and thus the orthology might be already supported by KOG annotation. Nevertheless, it will become more informative if basal plants (mosses, Lycophyte, and Amborella) orthologs were included in the trees as outgroups.

Besides, when writing this comment, I found invalid UniProt ID for Arabidopsis DFR, P13114. This ID is for CHS. Please check the tree OTU and its ID again.

Line 370 "The same conserved amino acid residues of FcCHS1, FcCHI1 and FcDFR1 as the corresponding genes of other vascular plants [55,56,59] indicate that they are truly the orthologs of known proteins in the three families.: This statement is not very appropriate. Just referring to some amino acid conservation can not tell orthology. Collective information with phylogenetic analysis with residual comparison can indicate their orthology. However, as mentioned above, DFR1 orthology seemed not enough supported. So depending on whether the authors reanalyze the tree or not, the sentence needs to be modified appropriately.

Fig. 2: New log-transformed values are fine, but the present values are "-"log10(p-value). Please change the legend.

Related to minor point 17, I still found misdescriptions of the company:
line 473 (Beckman coulter, Shanghai, China) -> US
line 540 (TIANGEN, Beijing, China) -> US, as the author revised in other parts

Some English-related concerns:
line 171~ (legend for fig4), 217~ (legend for fig5) "Multiply sequences alignment": "multiple sequence alignment" is more popular. Although "multiply alignment" was occasionally used, I feel it weird because "multiply" is generally a verb. It depends on the authors. In some parts of the manuscript, it is "multiple." 
line 391: "highly identity" -> "high identity"

I also have just a comment on the response to Minor Point 9, upon which the authors do not need to revise the manuscript:
This explanation is not convincing for me. First of all, as the transcriptome is transcribed from the genome, the ultimate goal of de novo transcriptome assembly is to get the same unigene number as the actual gene number on the genome. So the large unigene number far from the expected gene number in the genome cannot be explained by that there are many kinds of genes expressed in the tissue.
The concern behind this comment is that: assuming that the fig genome has 30K genes (like many diploid plants), but 180K unigene was assembled, it would be supposed that the assembly includes many fragmentarily assembled genes, contaminated sequences, allelic transcripts identified as unigenes, and so on. Collectively, the quality of the assembly would be bad. However, if the plant has a large genome via genome duplications and has, e.g.,120K genes bona fide, it is not surprising to obtain such a large number of unigenes. Concerning that only RNA extracted from a specific tissue, peel, was used for the assembly, it was natural to get less qualified assembly with quite a large number of unigene, because not all the genes were expected to be expressed in the peel. Interestingly, however, the assembly quality in this article looked very good in the sense of N50 length. So I felt it interesting, and this is why I recommended qualification using the other criteria, such as BUSCO, in comment 10 and wondered whether fig experienced genome duplication of not.
In the case of Camellia sinensis, the genome sequencing revealed that it has 34K-37K coding genes (https://doi.org/10.1073/pnas.1719622115). So the transcriptome assembly with 180K unigenes in Li et al. 2016 is supposed to include many undesirable sequences, even if considering the presence of non-coding transcripts. This may cause some problems in downstream analysis.

Author Response

Response to Reviewer 1 Comments

Point 1: Related to the response to minor point 9 (Figure 5):
Fig. 5E tree still does not support orthology of the fig gene, because the authors included just additional orthologous genes, but informative "outgroup" genes. From the topology of this unrooted tree, there is still a possibility that the clade of Fc and Mn genes is clustered with the outer sister family. To show the clear orthology of these genes to other known genes, I have commented that the authors need to add appropriate "outgroup genes" to root the tree. The outgroup can be either non-orthologous homolog genes, which are the closest sister of this gene family, or basal plant's orthologs.
For example, outgroup for DFR can be Arabidopsis BEN1(AT2G45400.1) and some orthologs of it.
The trees of CHS and CHI are also the same case. However, they are likely quite old genes, and thus the orthology might be already supported by KOG annotation. Nevertheless, it will become more informative if basal plants (mosses, Lycophyte, and Amborella) orthologs were included in the trees as outgroups.

Besides, when writing this comment, I found invalid UniProt ID for Arabidopsis DFR, P13114. This ID is for CHS. Please check the tree OTU and its ID again.

Response: Thank you for your suggestion. We have added Arabidopsis BEN1 (AT2G45400.1) and some orthologs of AtBEN1 as "outgroup genes" of FcDFR1 (Figure 5E, Page 6), and correspondingly revised the result in Result 2.6 and the Discussion 3.2.

Result 2.6:

"In the phylogenetic tree, BEN1 proteins (BRI1-5 ENHANCED 1, DFR-like protein), as the outgroup, verified that FcDFR1 are typical DFRs. Besides, FcDFR1 and MnDFR were in the same clade (Figure 5E)" (Line 210-212).

Discussion 3.2:

"FcCHS1, FcCHI1 and FcDFR1 composed the same conserved amino acid residues as the corresponding genes of other vascular plants [55,56,59] and their phylogenetic analysis support their close relationship with CHSs, CHIs and DFRs of other species. These results indicate that they are truly the ortholog of known proteins in the three families" (Line 369-373).

We have added CHSs and CHIs form mosses, Lycophyte, and Amborella as "outgroup genes" of FcCHS1 and FcCHI1 to root the tree (Figure 5D, F, Page 6), and correspondingly revised the result in Result 2.6 and the Discussion 3.2.

Result 2.6:

"Analysis of phylogenetic tree showed that CHSs were clustered into three group, where FcCHS1, MnCHS1 and MnCHS2 (Morus notabilis) were in the same branch. Also, FcCHS1 showed a close relation with those CHSs from the Rosaceae plants (Figure 5D). Those CHSs proteins of mosses and Lycophyte were acted as outgroup, confirmed FcCHS1 are orthologs of known CHS proteins" (Line 192-196).

"A phylogenetic tree created by the neighbour-joining method showed that FcCHI1 protein was closely related to MnCHI of Morus notabilis (Figure 5F). Moreover, orthologs from liverworts rooted FcCHI1 to CHIs family" (Line 201-203).

Discussion 3.2 is the same with DFRs.

We are sorry for our miswriting. We have checked the tree OTU and its ID, and corrected that wrong ones.

Point 2: Line 370 "The same conserved amino acid residues of FcCHS1, FcCHI1 and FcDFR1 as the corresponding genes of other vascular plants [55,56,59] indicate that they are truly the orthologs of known proteins in the three families.: This statement is not very appropriate. Just referring to some amino acid conservation can not tell orthology. Collective information with phylogenetic analysis with residual comparison can indicate their orthology. However, as mentioned above, DFR1 orthology seemed not enough supported. So depending on whether the authors reanalyze the tree or not, the sentence needs to be modified appropriately.

Response: We have revised the phylogenetic tree of DFRs according to your suggestion, and changed the sentence "The same conserved amino acid residues of FcCHS1, FcCHI1 and FcDFR1 as the corresponding genes of other vascular plants [55,56,59] indicate that they are truly the orthologs of known proteins in the three families" to "FcCHS1, FcCHI1 and FcDFR1 composed the same conserved amino acid residues as the corresponding genes of other vascular plants [55,56,59] and their phylogenetic analysis support their close relationship with CHSs, CHIs and DFRs of other species. These results indicate that they are truly the ortholog of known proteins in the three families" (Line 369-373).

Point 3: Fig. 2: New log-transformed values are fine, but the present values are "-"log10(p-value). Please change the legend.

Response: We have revised the legend of Figure 2B (-log10(p-value)) (Page 3).

Point 4: Related to minor point 17, I still found misdescriptions of the company:
line 473 (Beckman coulter, Shanghai, China) -> US
line 540 (TIANGEN, Beijing, China) -> US, as the author revised in other parts.

Response: We are sorry for our miswriting.

We have rechecked that the head company of TIANGEN and found it is located in Beijing. Therefore, we revised (TIANGEN, Atlanta, US) to (TIANGEN, Beijing, China). (Line 470).

We also revised (Beckman coulter, Shanghai, China) to (Beckman coulter, California, US) (Line 473).

Point 5:.Some English-related concerns:
line 171~ (legend for fig4), 217~ (legend for fig5) "Multiply sequences alignment": "multiple sequence alignment" is more popular. Although "multiply alignment" was occasionally used, I feel it weird because "multiply" is generally a verb. It depends on the authors. In some parts of the manuscript, it is "multiple." 
line 391: "highly identity" -> "high identity"

Response: Thanks for your kindly suggestion.

We have revised "Multiply sequences alignment" to "Multiple sequences alignment" in the legend of Figure 4 and Figure 5 (Line 172, 214, 215, 216).

We have revised "highly identity " to "high identity " (Line 391).

Point 6: I also have just a comment on the response to Minor Point 9, upon which the authors do not need to revise the manuscript:
This explanation is not convincing for me. First of all, as the transcriptome is transcribed from the genome, the ultimate goal of de novo transcriptome assembly is to get the same unigene number as the actual gene number on the genome. So the large unigene number far from the expected gene number in the genome cannot be explained by that there are many kinds of genes expressed in the tissue.
The concern behind this comment is that: assuming that the fig genome has 30K genes (like many diploid plants), but 180K unigene was assembled, it would be supposed that the assembly includes many fragmentarily assembled genes, contaminated sequences, allelic transcripts identified as unigenes, and so on. Collectively, the quality of the assembly would be bad. However, if the plant has a large genome via genome duplications and has, e.g.,120K genes bona fide, it is not surprising to obtain such a large number of unigenes. Concerning that only RNA extracted from a specific tissue, peel, was used for the assembly, it was natural to get less qualified assembly with quite a large number of unigene, because not all the genes were expected to be expressed in the peel. Interestingly, however, the assembly quality in this article looked very good in the sense of N50 length. So I felt it interesting, and this is why I recommended qualification using the other criteria, such as BUSCO, in comment 10 and wondered whether fig experienced genome duplication of not.
In the case of Camellia sinensis, the genome sequencing revealed that it has 34K-37K coding genes (https://doi.org/10.1073/pnas.1719622115). So the transcriptome assembly with 180K unigenes in Li et al. 2016 is supposed to include many undesirable sequences, even if considering the presence of non-coding transcripts. This may cause some problems in downstream analysis.

Response: Thank you for your detailed explanation. We will pay attention to the number of unigenes and try to explain this phenomenon in our future transcriptome analysis.
